# Domain Expansion: A Latent Space Construction Framework for Multi-Task Learning

**Chi-Yao Huang**[*,1]**, Khoa Vo**[*,1]**, Aayush Atul Verma**[*,1]**, Duo Lu**[2]**, Yezhou Yang**[1]

[1]Arizona State University    [2]Rider University
{cy.huang, ngocbach, averma90, yz.yang}@asu.edu
{dlu@rider.edu}
[*] Equal contribution

## Abstract

Training a single network with multiple objectives often leads to conflicting gradients that degrade shared representations, forcing them into a compromised state that is suboptimal for any single task—a problem we term latent representation collapse. We introduce Domain Expansion, a framework that prevents these conflicts by restructuring the latent space itself. Our framework uses a novel orthogonal pooling mechanism to construct a latent space where each objective is assigned to a mutually orthogonal subspace. We validate our approach across diverse benchmarks—including ShapeNet, MPIIGaze, and Rotated MNIST—on challenging multi-objective problems combining classification with pose and gaze estimation. Our experiments demonstrate that this structure not only prevents collapse but also yields an explicit, interpretable, and compositional latent space where concepts can be directly manipulated.

## 1 Introduction

Representation learning seeks to map raw data into meaningful latent features in latent space, a principle that underpins successes across machine learning, from classification (Chen et al., 2020; Khosla et al., 2020) to large-scale multimodal learning (Radford et al., 2021; Liu et al., 2023b). A common paradigm is to train a single, unified network to satisfy multiple learning objectives simul-

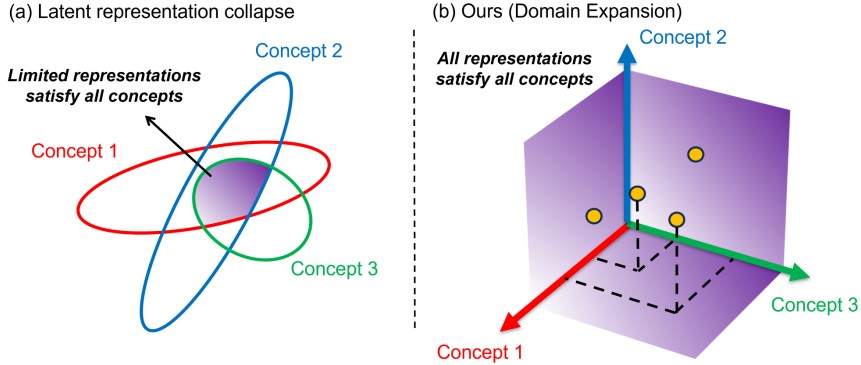

Figure 1: **(a) Latent representation collapse.** In standard multi-task learning, competing objectives lead to latent representation collapse, where the solution spaces for different concepts (colored ellipses) overlap in only a small, compromised region. **(b) Domain Expansion.** In contrast, our method assigns each concept to an orthogonal basis vector in the latent space, preventing interference and creating a structured, interpretable representation where features for each concept are clearly separated.

taneously—for example, performing both classification and regression on the same input. However, this approach exposes a fundamental challenge: competing objectives often produce conflicting gradients, pulling shared latent features in opposing directions. This interference can degrade model performance, a phenomenon we formalize as *latent representation collapse* (Fig. 1(a)).

When collapse occurs, the network carves out only a small, compromised region of the latent space that partially satisfies all objectives, failing to excel at any. This not only limits predictive accuracy but also leads to entangled, uninterpretable representations where the underlying factors of variation are obscured. While existing multi-task learning methods attempt to mitigate this by re-weighting task losses or projecting gradients at runtime (Yu et al., 2020; Sener & Koltun, 2018), these approaches act on the optimization process itself, not the structure of the latent space. The core problem of designing a representation space that is inherently robust to interference remains underexplored.

To address this gap, we propose Domain Expansion, a new framework for constructing latent spaces that are multi-objective by design. Instead of mediating gradient conflicts, we prevent them structurally. The core of our framework is orthogonal pooling, a lightweight architectural primitive that constrains features for different objectives to lie in mutually orthogonal subspaces. This enforcement of orthogonality ensures that learning one objective cannot interfere with another, allowing the model to learn potent, disentangled features for all tasks simultaneously (Fig. 1(b)).

We validate Domain Expansion on a diverse set of benchmarks, including ShapeNet (Chang et al., 2015), MPIIGaze (Zhang et al., 2019), and rotated MNIST (Deng, 2012), tackling multi-objective problems that combine classification with various forms of regression like pose and gaze estimation. Across all experiments, our method successfully resolves latent representation collapse. Furthermore, we demonstrate that the resulting latent space is not a black box; it possesses an explicit, interpretable structure where distinct objectives are disentangled by design, enabling analysis of their relationships. Our contributions are:

- We formalize latent representation collapse, a critical failure mode in multi-objective representation learning.
- We introduce Domain Expansion, a framework that uses orthogonal pooling to construct a latent space with mutually orthogonal subspaces, preventing task interference by design.
- We demonstrate that our method constructs an explicit and interpretable latent space, where orthogonal axes correspond to distinct concepts, enabling compositional inference and analysis.

## 2 RELATED WORK

The goal of representation learning is to build latent spaces that capture meaningful factors of variation from raw data. Modern methods often learn such representations by optimizing for multiple, sometimes implicit, objectives. For instance, contrastive learning (Chen et al., 2020; He et al., 2019; Khosla et al., 2020) structures the latent space by pulling similar samples together while pushing dissimilar ones apart. Large multimodal models like CLIP (Radford et al., 2021) learn powerful, transferable features by aligning representations from different modalities, such as images and text. The success of these models hinges on their ability to create a unified representation that satisfies these diverse learning signals.

This principle is formalized in multi-task learning (MTL), which aims to improve generalization and efficiency by training a single model on multiple tasks simultaneously (Zamir et al., 2018). However, a central challenge in MTL is negative transfer (Ruder, 2017), where optimizing for one task degrades the performance of another. This issue often arises from conflicting gradients, where different task-specific losses pull the shared network parameters in opposing directions. This conflict can lead to the phenomenon we term latent representation collapse, where the learned features are a poor compromise that fails to adequately solve any single task.

A dominant line of work addresses this challenge by manipulating task gradients during the optimization process. These methods aim to find a less conflicting update vector by altering the gradients from individual tasks. For example, GradNorm (Chen et al., 2017) dynamically adjusts the weights of each task's loss to balance their training rates. Other methods focus on the geometry of the

Figure 2: **Problem statement.** (Left) Real-world inputs contain multiple concepts simultaneously. (Center) Standard multi-objective training leads to *latent representation collapse*, where concepts interfere and the latent space becomes entangled. (Right) Our *Domain Expansion* resolves this by assigning each concept to a mutually orthogonal subspace, yielding an explicit, interpretable, and compositional latent space.

gradients themselves. PCGrad (Yu et al., 2020) and IMTL (Liu et al., 2021b) identify conflicting gradients and projects them onto the normal plane of others to remove the conflicting components. Building on this, methods like CAGrad (Liu et al., 2021a) and MGDA (Sener & Koltun, 2018) seek a common gradient that represents a Pareto optimal solution or minimizes the worst-case loss across tasks.

While effective, these gradient-level methods are fundamentally reactive; they resolve conflicts at each training step after they have already occurred. In contrast, our work offers a proactive, representation-centric solution. Rather than mediating conflicts during optimization, Domain Expansion constructs a latent space with inherent structural properties that prevent interference by design. Through its orthogonal pooling, our framework enforces that different objectives operate in separate, non-interfering subspaces. This proactive approach eliminates the need for runtime gradient manipulation and results in a latent space that is more explicit, interpretable, and robust to multi-objective training.

## 3 METHOD

Our goal is to design a latent space where a single representation can effectively encode information for multiple, competing objectives without interference. This section details our approach in three parts. First, we formalize the problem of latent representation collapse. Second, we introduce our solution, a framework we call Domain Expansion. Finally, we demonstrate how this method yields a structured and compositional latent space with powerful algebraic properties.

### 3.1 PROBLEM STATEMENT

A standard representation learning pipeline involves an encoder, $Enc(\cdot)$, that maps an input $z$ from the data space $\mathcal{Z}$ to a latent representation $f \in \mathcal{F} \subset \mathbb{R}^D$. For each learning objective $m$, a corresponding decoder, $Dec(\cdot)$, then maps the latent representation $f$ to a target concept $\mathcal{C}$. The quality of this mapping is measured by an objective loss $\mathcal{L}$. This can be expressed as:

$$\mathcal{Z} \xrightarrow{\text{Enc}} \mathcal{F} \xrightarrow{\text{Dec}} \mathcal{C} \quad \text{(evaluated by } \mathcal{L}\text{)} \tag{1}$$

In representation learning, the objective loss $\mathcal{L}$ is often defined directly on the latent representations $\mathcal{F}$. The loss function's goal is to impose a desired structure on the latent space (e.g., clustering or ranking), guided by the supervision from the target concept $\mathcal{C}$.

The challenge arises when this framework is extended to handle a set of $M$ objectives simultaneously. In this scenario, a single, shared representation $\mathcal{F}$ is subjected to multiple objective losses, $\{\mathcal{L}_m\}_{m=0}^{M-1}$, each guided by a distinct target concept, $\{\mathcal{C}_m\}_{m=0}^{M-1}$. The total loss becomes a weighted sum applied to these shared representations:

$$\mathcal{L}_{\text{total}} = \sum_{m \in M} w_m \cdot \mathcal{L}_m(\mathcal{F}, \mathcal{C}_m). \tag{2}$$

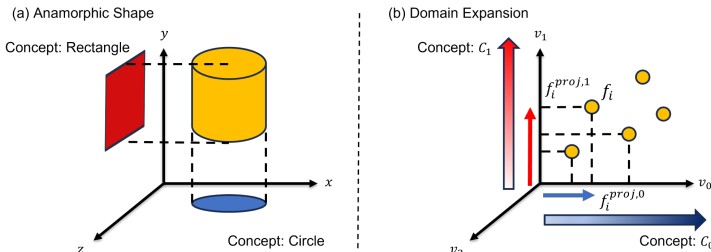

Figure 3: A single latent vector represents multiple concepts through orthogonal projections, inspired by anamorphic art. (a) An anamorphic object, such as a cylinder, reveals different primitive concepts (a circle vs. a rectangle) when viewed from orthogonal directions. (b) Analogously, our method treats a single latent feature as a rich object that encodes multiple concepts simultaneously. The specific value for each concept is determined by its projection onto a corresponding orthogonal axis in the latent space.

As we have argued, naively minimizing this sum often leads to latent representation collapse, where conflicting structural demands from different losses force the shared representation $\mathcal{F}$ into a compromised state that is suboptimal for all objectives (see Fig. 2).

## 3.2 DOMAIN EXPANSION

To prevent latent collapse in multi-objective learning, we introduce Domain Expansion, a framework that structures the latent space $\mathcal{F}$ into a set of dedicated, orthogonal subspaces. The core idea is to ensure that the updates for each objective $m$ are confined to a unique subspace, preventing them from interfering with the representations of other objectives (see Fig. 3). Our key insight is that the principal directions of variance in the latent space—its eigenvectors—can be harnessed to serve as an orthonormal basis for these dedicated subspaces.

The framework is a dynamic process applied at each training epoch, consisting of three steps:

1. **Find Principal Axes.** First, we estimate the empirical mean $\mu$ and covariance $\Sigma$ of the latent feature distribution over the current batch or entire training set:

$$\mu = \mathbb{E}_{z \sim p(z)}[Enc(z)], \qquad \Sigma = \mathbb{E}_{z \sim p(z)}[(Enc(z) - \mu)(Enc(z) - \mu)^\top]. \tag{3}$$

We then perform an eigendecomposition of the covariance matrix to find an orthonormal basis of eigenvectors, $V = [v_0, v_1, \ldots, v_{D-1}]$.

$$\Sigma = V\Lambda V^\top, \qquad V^\top V = I. \tag{4}$$

2. **Define the Orthogonal Domain.** We select the top $M$ eigenvectors (those with the largest eigenvalues) to form our conceptual basis, which we call the *domain*, $V_M = \{v_m \mid m \in M\}$. Each eigenvector $v_m$ is assigned then to represent a single target concept $\mathcal{C}_m$. This assignment ensures that the 1D subspace spanned by $v_m$ becomes exclusive for all projected features related to that concept. From this, we define the set of orthogonal subspaces and their corresponding projection operators:

$$\mathcal{F}_m^{\text{proj}} = \text{span}(v_m), \quad \text{Proj}_m = v_m v_m^\top, \quad \forall m \in M. \tag{5}$$

3. **Orthogonal Pooling.** After defining the orthogonal domain, we decompose the latent feature $f$ into each orthogonal, concept-specific subspace via computing its projection onto each of the domain's axes:

$$f^{\text{proj},m} = \text{Proj}_m(f - \mu), \quad \forall m \in M. \tag{6}$$

This step, which we term *orthogonal pooling*, reformulates the learning pipeline as a one-to-many mapping from the shared space to the set of orthogonal subspaces, where each

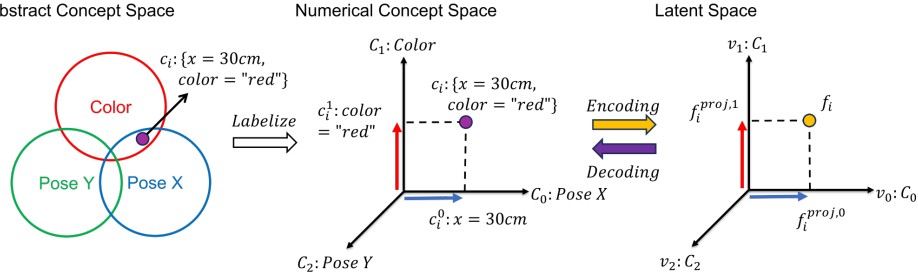

Figure 4: **Concept space vs. latent space:** (Left) Real-world inputs possess multiple abstract concepts simultaneously, such as color and pose. (Center) For training, we define a numerical concept space by assigning coordinates to these attributes. (Right) Our model then learns a mapping between this numerical space and its own internal, orthogonal latent space. This process creates an explicit structure where each axis corresponds to a single concept, allowing the model to robustly represent and manipulate the independent attributes of the input.

subspace corresponds to a single target concept.

$$
\mathcal{Z} \xrightarrow{\text{Enc}} \mathcal{F} \xrightarrow{\text{Orthogonal Pooling}}
\begin{cases}
\mathcal{F}_0^{\text{proj}} \xrightarrow{\text{Dec}_0} \mathcal{C}_0 \\
\mathcal{F}_1^{\text{proj}} \xrightarrow{\text{Dec}_1} \mathcal{C}_1 \\
\quad\vdots \\
\mathcal{F}_{M-1}^{\text{proj}} \xrightarrow{\text{Dec}_{M-1}} \mathcal{C}_{M-1}
\end{cases}
\tag{7}
$$

The total training loss is then computed as the sum of individual losses on these independent, projected latent features:

$$
\mathcal{L}_{\text{total}} = \sum_{m \in M} w_m \cdot \mathcal{L}_m(\mathcal{F}_m^{\text{proj}}, \mathcal{C}_m).
\tag{8}
$$

By decomposing the latent space into a set of dedicated, orthogonal subspaces for each concept, our framework ensures that the resulting loss gradients are inherently decoupled, preventing representation collapse by design.

## 3.3 PROPERTIES AND OPERATORS OF THE DOMAIN

The orthogonal structure of our method is not merely a training aid; it endows the latent space with powerful properties, turning it into an interpretable and compositional *concept algebra*.

To formalize this, we define our terminology in order:

1. A *target concept* $\mathcal{C}_m$ is the set of all possible values for a given attribute (e.g., Pose $X$).
2. A *target instantiated concept* $c^m$ is a single value from one such set, where $c^m \in \mathcal{C}_m$ (e.g., x = 30 cm).
3. The *concept space* $\mathcal{C}_{\text{space}}$ is the set of all possible combinations of attributes, formally defined as the Cartesian Product of the target concept sets (i.e. pose, color, category, etc.):

$$
\mathcal{C}_{\text{space}} = \mathcal{C}_0 \times \mathcal{C}_1 \times \cdots \times \mathcal{C}_{M-1} = \overset{M-1}{\underset{m=0}{\times}} \mathcal{C}_m.
\tag{9}
$$

4. Finally, an *instantiated concept* $c$ is a single element from this space, represented as a tuple $c = \{c^0, c^1, \ldots, c^{M-1}\}$. During training, each input $z_i$ is associated with a specific instantiated concept $c_i$.

For each target instantiated concept $c^m$ of an instantiated concept, we assume a mapping to a latent projection $f^{\text{proj},m}$ via the decoder $Dec_m$. (The implementation of the decoder, $Dec_m$ and $Dec_m^{-1}$, is detailed in Appendix A.1).

$$
c_i^m = \text{Dec}_m(f_i^{\text{proj},m}), \quad f_i^{\text{proj},m} = \text{Dec}_m^{-1}(c_i^m)
\tag{10}
$$

**Property 1: Orthogonality of Target Concepts.** Because each projection subspace $\mathcal{F}_m^{\text{proj}}$ is orthogonal to the others, the target concepts they represent are disentangled in the latent space:

$$\mathcal{F}_0^{\text{proj}} \perp \mathcal{F}_1^{\text{proj}} \perp \cdots \perp \mathcal{F}_{M-1}^{\text{proj}} \implies \mathcal{C}_0 \perp \mathcal{C}_1 \perp \cdots \perp \mathcal{C}_{M-1}. \tag{11}$$

**Property 2: Multi-concept Encoding.** A single latent feature $f_i$ simultaneously encodes a full instantiated concept $c_i$. The feature can be decomposed into its orthogonal projections, which are then decoded into their corresponding target instantiated concepts (see Fig. 4):

$$f_i \xrightarrow{\text{Pooling}} \{f_i^{\text{proj},0}, \ldots, f_i^{\text{proj},M-1}\} \xrightarrow{\text{Dec}} \{c_i^0, \ldots, c_i^{M-1}\} \to c_i. \tag{12}$$

Conversely, the full latent feature can be reconstructed from its components:

$$c_i \to \{c_i^0, \ldots, c_i^{M-1}\} \xrightarrow{\text{Dec}^{-1}} \{f_i^{\text{proj},0}, \ldots, f_i^{\text{proj},M-1}\} \xrightarrow{\text{Reconst}} f_i, \tag{13}$$

where the reconstruction is defined as:

$$f_i = \mu + \sum_{m \in M} f_i^{\text{proj},m}. \tag{14}$$

**Operator 1: Concept-Specific Adjustment ($\oplus^m$) and ($\ominus^m$).** This operator adjusts an instantiated concept $c_i$ by applying a change defined by a single target instantiated concept $c_\Delta^m \in \mathcal{C}_m$. The operation modifies the latent feature $f_i$ without affecting any other target concepts. First, we find the latent vector for the change, $f_\Delta^{\text{proj},m} = Dec_m^{-1}(c_\Delta^m)$. The adjusted latent feature is then given by simple vector addition or subtraction:

$$f_j = f_i \pm f_\Delta^{\text{proj},m}. \tag{15}$$

The full derivation showing the correspondence between the concept and latent spaces is as follows:

$$c_i \oplus^m c_\Delta^m \to \{c_i^0, \ldots, \{c_i^m \oplus^m c_\Delta^m\}, \ldots c_i^{M-1}\} \tag{16}$$

$$\xrightarrow{\text{Dec}^{-1}} \{f_i^{\text{proj},0}, \ldots, \{f_i^{\text{proj},m} + f_\Delta^{\text{proj},m}\}, \ldots, f_i^{\text{proj},M-1}\} \tag{17}$$

$$\xrightarrow{\text{Reconst}} f_i + f_\Delta^{\text{proj},m}. \tag{18}$$

The derivation for the subtraction operator ($\ominus^m$) is analogous.

**Operator 2: Concept Composition ($\oplus$) and ($\ominus$).** This operator composes two full instantiated concepts, $c_p$ and $c_q$, by operating on their corresponding latent representations, $f_p$ and $f_q$. The composition is achieved through simple vector addition or subtraction:

$$f_{pq} = f_p \pm f_q. \tag{19}$$

This operation corresponds to a component-wise combination in each orthogonal subspace:

$$c_p \oplus c_q \to \{\{c_p^0 \oplus^0 c_q^0\}, \{c_p^1 \oplus^1 c_q^1\}, \ldots, \{c_p^{M-1} \oplus^{M-1} c_q^{M-1}\}\} \tag{20}$$

$$\xrightarrow{\text{Dec}^{-1}} \{\{f_p^{proj,0} + f_q^{proj,0}\}, \{f_p^{proj,1} + f_q^{proj,1}\}, \ldots, \{f_p^{proj,M-1} + f_q^{proj,M-1}\}\} \tag{21}$$

$$\xrightarrow{\text{Reconst}} f_p + f_q. \tag{22}$$

The derivation for the subtraction operator ($\ominus$) is analogous.

## 4 EXPERIMENTS

To validate our framework, we formulated and tested three key hypotheses (H):

- H1: Does training a single network with multiple objectives lead to **latent representation collapse** as expected?
- H2: Does **Domain Expansion** prevent this collapse and outperform standard multi-task learning baselines?
- H3: Does our method create a truly **compositional and inferable** latent space, as claimed?

Through designing experiments to test these hypotheses, we aimed to systematically evaluate our framework's effectiveness and validate its core assumptions. Specifically, we conduct experiments on the ShapeNet dataset (Chang et al., 2015), a standard benchmark for 3D object classification and pose estimation. We further validate our method on the MPIIGaze (Zhang et al., 2019) and Rotated MNIST (Deng, 2012) datasets (see Appendix A.3 for full details).

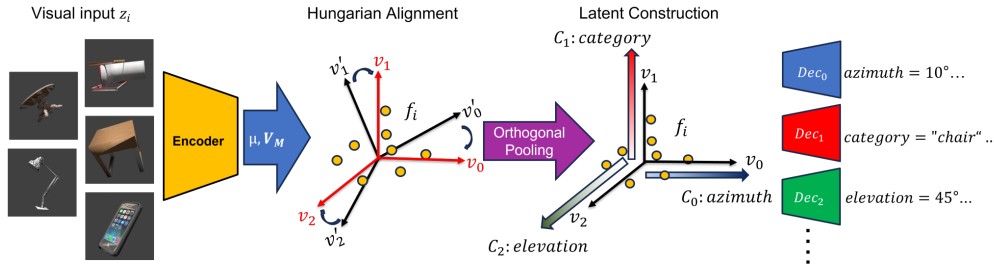

Figure 5: **System overview:** An input image is first passed through an encoder to produce a latent feature. From the distribution of these features across the dataset, we compute the mean ($\mu$) and an orthogonal basis of eigenvectors ($V_M$). To ensure consistency during training, this basis is stabilized across epochs using Hungarian alignment. The orthogonal pooling then projects each latent feature onto these basis vectors. Finally, these projected, non-interfering representations are fed into concept-specific decoders to produce the final outputs.

## 4.1 EXPERIMENTAL SETUP

**Architecture and Training.** We adopt an encoder-decoder architecture here. The encoder, $Enc(\cdot)$, is a ResNet-50 (He et al., 2015) backbone that produces a 2048-dimensional latent feature, $f \in \mathcal{F}$. For each of the $M$ objectives, its corresponding decoder, $Dec_m(\cdot)$, is a single linear layer. Training proceeds in two stages: first, we train the encoder while dynamically updating the orthogonal basis ($V_M$ and $\mu$) at each epoch. To stabilize the basis during early training, we align the eigenvectors between epochs using the Hungarian algorithm (Kuhn, 1955) (see Appendix A.2). Second, once the latent space has converged, we freeze the encoder and train the simple linear decoders on the final, fixed representations (see Fig. 5).

**Dataset and Concepts.** We use the ShapeNet dataset, rendering approximately 30,000 images from 10 models per category using the 3D-R2N2 pipeline (Choy et al., 2016). We define five target concepts: three for regression—azimuth (sampled from $[-\pi/2, \pi/2]$), elevation, and in-plane rotation (both sampled from $[-\pi/4, \pi/4]$)—and two for classification (object category and model ID). We denote these concepts as $\{\mathcal{C}_{az}, \mathcal{C}_{el}, \mathcal{C}_{rot}, \mathcal{C}_{cat}, \mathcal{C}_{id}\}$.

**Loss Functions.** To structure the latent space, we use a weighted combination of two representation learning losses that operate on the high-dimensional latent vectors. Supervised Contrastive (SupCon) (Khosla et al., 2020) imposes a clustering constraint using a binary mechanism (positive vs. negative pairs). We adapt the standard SupCon loss by replacing the inner product similarity with L2 distance, which is more compatible with our projection-based method. Rank-N-Contrast (RNC) (Zha et al., 2023) enforces a more fine-grained ranking mechanism. For both losses, we set the temperature $\tau = 2.0$. The final loss weights $w_m$ are to 1.0 for RNC and 0.02 for SupCon.

Based on these components, we define two primary sets of objectives for our experiments, which correspond in order to the concepts $\{\mathcal{C}_{az}, \mathcal{C}_{el}, \mathcal{C}_{rot}, \mathcal{C}_{cat}, \mathcal{C}_{id}\}$. **Objective Set 1** uses the RNC loss for all five concepts. **Objective Set 2** uses RNC for the first three concepts (regression) and SupCon for the final two (classification).

## 4.2 EVALUATING MULTI-OBJECTIVE PERFORMANCE

To answer our first two research questions (H1 and H2), we evaluate our method against a suite of baselines. We establish a simple baseline trained with a weighted sum of all losses (Eq. 2). We also compare against three gradient-based MTL methods designed to mitigate gradient conflict: Nash-MTL (Navon et al., 2022), FAMO (Liu et al., 2023a), and IMTL (Liu et al., 2021b). Finally, we evaluate the performance of our full proposed method, Domain Expansion (Ours).

We evaluate the quality of the learned representations for each concept using metrics tailored to the task type. For regression concepts (azimuth, elevation, rotation), we measure the Spearman's rank correlation (Spearman, 1987). For classification concepts (category, ID), we measure the V-measure

Table 1: Comprehensive comparison of representation quality, predictive performance, and concept composition. Arrows indicate whether higher (↑) or lower (↓) values are better.

| Objective Set | Method | Representation & Predictive Performance | | | | | | | | | | Concept Comp. |
| | | Spearman ↑ | | | V-score ↑ | | MAE° ↓ | | | Acc. ↑ | | Sim. ↑ |
| | | az | el | rot | cat | id | az | el | rot | cat | id | $\oplus$ and $\ominus$ |
|---|---|---|---|---|---|---|---|---|---|---|---|---|
| | baseline | 0.41 | 0.34 | 0.35 | 0.16 | 0.14 | 0.12 | 0.09 | 0.09 | 0.28 | 0.37 | 0.22 |
| | FAMO | 0.49 | 0.41 | 0.42 | 0.00 | 0.00 | 0.12 | 0.09 | 0.09 | 0.19 | 0.18 | 0.28 |
| Objective Set 1 | Nash-MTL | 0.38 | 0.41 | 0.42 | 0.00 | 0.00 | 0.11 | 0.09 | 0.09 | 0.17 | 0.13 | 0.28 |
| | IMTL | 0.31 | 0.16 | 0.16 | 0.39 | 0.28 | 0.14 | 0.11 | 0.12 | 0.92 | 0.79 | 0.14 |
| | **Ours** | 0.95 | 0.87 | 0.85 | 0.99 | 0.91 | 0.08 | 0.08 | 0.09 | 0.99 | 0.97 | 0.95 |
| | baseline | 0.01 | 0.01 | 0.01 | 0.99 | 0.00 | 0.77 | 0.38 | 0.38 | 0.99 | 0.99 | 0.42 |
| | FAMO | 0.28 | 0.23 | 0.22 | 0.99 | 0.00 | 0.19 | 0.14 | 0.13 | 0.99 | 0.99 | 0.28 |
| Objective Set 2 | Nash-MTL | 0.45 | 0.39 | 0.39 | 0.15 | 0.00 | 0.12 | 0.08 | 0.09 | 0.99 | 0.99 | 0.35 |
| | IMTL | 0.39 | 0.18 | 0.16 | 0.99 | 0.00 | 0.15 | 0.11 | 0.13 | 0.99 | 0.99 | 0.28 |
| | **Ours** | 0.95 | 0.87 | 0.85 | 0.98 | 0.96 | 0.07 | 0.08 | 0.09 | 0.98 | 0.94 | 0.93 |

(V-score) (Rosenberg & Hirschberg, 2007) to evaluate cluster quality. We also report standard predictive metrics: Mean Absolute Error (MAE°) for regression and accuracy (Acc.) for classification.

The results in Table 1 demonstrate the effectiveness of our Domain Expansion framework in preventing latent representation collapse. The baseline model confirms that naive multi-objective training leads to a disorganized latent space, exhibiting poor performance on representation quality metrics like Spearman correlation and V-score. A particularly revealing result is visible for Objective Set 2, where several baselines achieve high classification accuracy but have a V-score near zero. This discrepancy highlights that they learn a superficial shortcut for the predictive task while their latent space remains collapsed.

In contrast, Domain Expansion significantly outperforms all baselines, achieving superior scores on both the representation metrics and the final predictive tasks. This quantitative success is mirrored in our qualitative visualizations in Fig. 6. Whereas the latent spaces of baseline methods appear entangled and unstructured, the space learned by our method is clearly organized, with concepts aligning along their corresponding orthogonal axes.

### 4.3 PROBING THE COMPOSITIONAL LATENT SPACE

Finally, we designed an experiment to test the **composition operator** ($\oplus$ and $\ominus$) and verify that our latent space is truly inferable (H3). The goal is to see if we can transform one latent vector $f_p$ into a target vector $f_q$ by applying a conceptual difference defined in the concept space.

1. We split the test set into two halves, P and Q. For a pair of samples $(z_p, c_p)$ and $(z_q, c_q)$, our goal is to synthetically reconstruct $f_q$.

2. The ground-truth target vector is obtained directly from the encoder: $f_q = \text{Enc}(z_q)$.

3. We define the conceptual difference between the two samples as $c_\Delta = c_q \ominus c_p$.

4. We then create a synthetic target concept by applying this difference to our source concept: $c_q^* = c_p \oplus c_\Delta$.

5. Using our framework's operators, we reconstruct the latent vector corresponding to this synthetic concept: $f_q^* = \text{Reconst}(c_q^*)$ (Eq. 13).

6. We evaluate the quality of this synthetic reconstruction by computing the average cosine similarity between the reconstructed vectors and the ground-truth vectors over all pairs in the test set Q: $\mathbb{E}_{(z_p, z_q)}[\cos(f_q, f_q^*)]$.

As shown in the final column of Table 1, our method's concept composition performance, measured by cosine similarity, is substantially higher than all baselines. This result confirms that our latent space is not a "black box," but rather a meaningful, compositional structure where conceptual operations correspond to simple vector arithmetic, enabling accurate inference and manipulation.

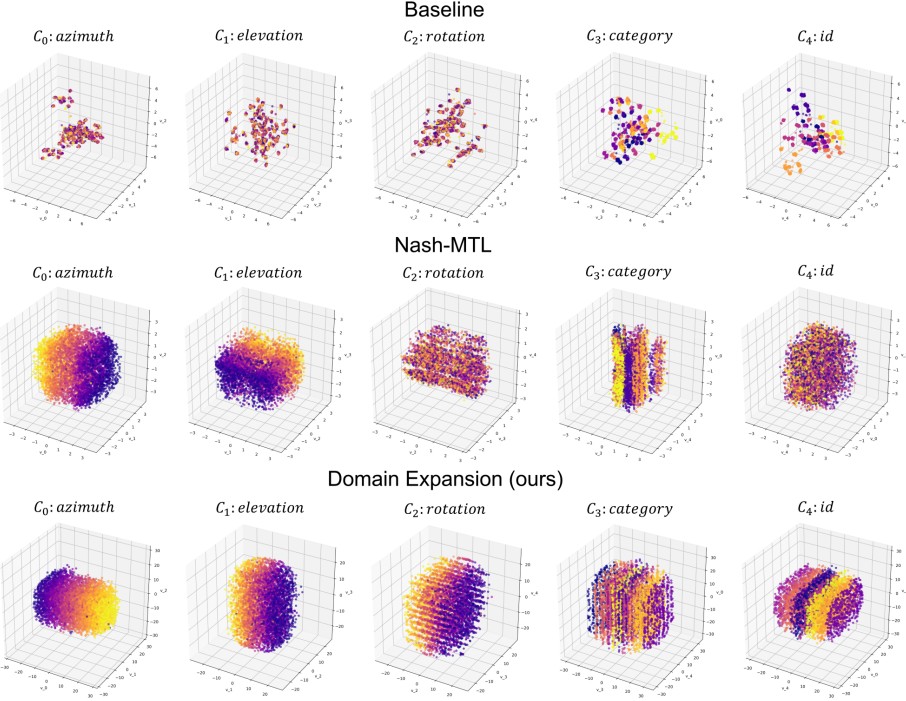

Figure 6: **Qualitative Comparison of Latent Space Structures.** We visualize the latent spaces from the baseline, Nash-MTL, and Ours (top to bottom rows) via PCA. Points are colored by their ground-truth concept value, from low (dark purple) to high (bright yellow). Our method (bottom row) demonstrates a distinctly more organized structure with clear directional alignment for each concept, unlike the entangled representations produced by the baseline methods.

## 5 DISCUSSION AND LIMITATIONS

A key design choice in our framework is leveraging projection-based supervision. Unlike other approaches that might constrain the entire latent vector, our method only guides its components along the pre-defined orthogonal axes. It is innately permissive by design: it provides the encoder with significantly more degrees of freedom to learn a rich internal representation, which we believe contributes to the performance and compositional properties we observed.

The primary strength of such a structured space is its support for high-level conceptual compositions that are difficult for human beings to understand intuitively. For instance, our framework can systematically represent a novel concept like "chair" $\oplus$ "boat" via the simple latent space operation $f_{chair} + f_{boat}$. Our current limitation, therefore, is not in representation but in decoding these abstract concepts. A promising direction for future work is to pair our encoder with a generative model, such as an LLM or diffusion model, to interpret these latent compositions into human-understandable outputs.

## 6 CONCLUSION

In this work, we introduced Domain Expansion, a framework that addresses latent representation collapse by constructing a latent space with dedicated orthogonal subspaces for each task. Our experiments demonstrated that this design alleviates the risk of task interference and yields an explicit and compositional latent space that supports algebraic concept manipulation. This approach suggests a future pathway for a more structured bridge between high-level concepts and a model's learned representation, laying a promising foundation for more controllable and interpretable models with applications in areas like algorithmic fairness and controllable multi-modal content generation.

## 7 ACKNOWLEDGMENT

This research is sponsored by NSF, the Partnerships for Innovation grant (#2329780) and TRINA Mothership project. We thank the Research Computing (RC) at Arizona State University (ASU) (Jennewein et al., 2023) and the NSF NAIRR program for their generous support in providing computing resources. The views and opinions of the authors expressed herein do not necessarily state or reflect those of the funding agencies and employers.

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
