## A APPENDIX

### A.1 ON THE INVERTIBILITY OF THE DECODER

In our method, the concept operators require a mapping from a concept instance back to its latent representation, denoted as $f^{\text{proj},m} = \text{Dec}_m^{-1}(c^m)$. This section details how we implement this inverse mapping.

Our decoders are single linear layers:

$$c^m = W \cdot f^{\text{proj},m} + b, \tag{23}$$

where $W$ and $b$ are the layer's weights and bias. Since the latent space dimension ($D = 2048$) is much larger than the concept dimension (e.g., 1 for a regression target or the number of classes (logit) for a classification target), the weight matrix $W$ is non-square and a standard inverse does not exist. The inverse problem is **ill-posed**: for any given output $c^m$, there is an entire affine subspace of possible inputs that could have produced it.

However, our Domain Expansion framework provides a powerful geometric constraint that makes this problem tractable. By construction, we know that any valid projected feature $f^{\text{proj},m}$ must lie on the 1D subspace spanned by its corresponding eigenvector, $v_m$. This constraint resolves the ambiguity of the inverse mapping.

Therefore, we can implement the inverse decoder as a two-step process, as illustrated in Figure 7. First, we find an arbitrary solution, $(f^{\text{proj},m})^*$, in the high-dimensional preimage (e.g., using the pseudo-inverse). Second, we simply project this arbitrary solution back onto the correct eigenvector to recover the unique, valid latent feature:

$$f^{\text{proj},m} = \text{Proj}_m((f^{\text{proj},m})^*). \tag{24}$$

This elegant solution is a direct benefit of the explicit structure imposed by our method.

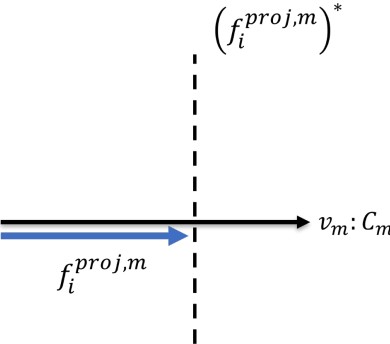

Figure 7: **Recovering a unique inverse for the decoder.** The set of all possible latent features that could produce a given output forms a high-dimensional preimage (illustrated by the dashed line). Our method provides a strong constraint: the true solution must lie on the subspace spanned by the eigenvector $v_m$. By projecting an arbitrary solution $(f^{\text{proj},m})^*$ onto this subspace, we can recover the unique, correct latent feature $f^{\text{proj},m}$.

### A.2 SUBSPACE STABILITY AND ALIGNMENT

During iterative training, the eigenvectors that define our conceptual subspaces can be subject to two forms of ambiguity. First, the order of the learned eigenvectors may permute between epochs. Second, an eigenvector $v$ and its negative $-v$ span the same 1D subspace. These ambiguities can destabilize training as the network may associate a concept with a different eigenvector at each step.

To resolve this, we enforce consistency across training epochs using the Hungarian algorithm. At the end of each epoch, we compute a pairwise cosine similarity matrix between the newly learned eigenvectors and the eigenvectors from the previous epoch. The Hungarian algorithm then finds the

optimal one-to-one assignment that maximizes the similarity, resolving any permutation ambiguity. We also align the sign of each eigenvector to ensure the cosine similarity is positive.

This alignment procedure ensures that the learned subspaces stabilize and converge. Figure 8 visualizes this process, showing the cosine similarity between aligned eigenvectors from consecutive epochs. The similarities approach 1.0, indicating convergence. Table 2 confirms this, showing that the final similarities are nearly perfect, which demonstrates that the conceptual subspaces are stable upon the completion of training.

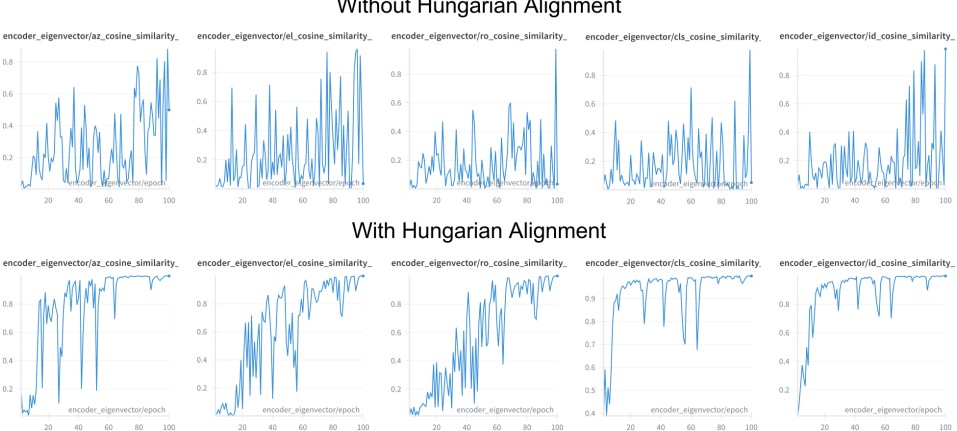

Figure 8: **Eigenvector stabilization during training.** We plot the cosine similarity between the aligned eigenvectors for each concept from the current and previous training epochs. The increasing similarity demonstrates the convergence of the learned subspaces. From left to right, the plots correspond to the subspaces for azimuth, elevation, in-plane rotation, category, and ID, respectively.

Table 2: Effect of Hungarian Alignment on Eigenvector Stability

| Method | Azim. | Elev. | Rot. | Cat. | ID |
|---|---|---|---|---|---|
| Without Hungarian | 0.49 | 0.04 | 0.03 | 0.05 | 0.98 |
| With Hungarian | 0.99 | 0.99 | 0.99 | 0.99 | 0.99 |

### A.3 EXPERIMENT RESULT OF OTHER DATASET

To demonstrate the generalizability of our Domain Expansion framework, we conduct experiments on two additional, diverse benchmarks: the MPIIGaze dataset for fine-grained gaze estimation and a custom Rotated MNIST dataset for classification under pose variation.

**MPIIGaze Setup.** The MPIIGaze dataset (Zhang et al., 2019) contains approximately 30,000 images from 15 participants for the task of gaze estimation. Following our setup for ShapeNet, we use a ResNet-50 backbone. We define four target concepts corresponding to the ordered set $\{\mathcal{C}_x, \mathcal{C}_y, \mathcal{C}_z, \mathcal{C}_{id}\}$: the three axes of 3D gaze direction (x,y,z) for regression, and the participant ID for classification.

Based on this ordering, we evaluate two objective sets. Objective Set 1 uses the RNC loss for all four concepts. Objective Set 2 uses RNC for the three regression concepts and SupCon for the classification concept. The complete results for these experiments, detailing representation quality, predictive performance, and concept composition, are summarized in Table 3.

**Rotated MNIST Setup.** To further test our model's ability to handle pose concepts, we created a Rotated MNIST dataset by applying random rotations in the range $[-\pi/2, \pi/2]$ to images from the original MNIST dataset (Deng, 2012). For this task, we use a ResNet-18 backbone and define two

concepts corresponding to the ordered set $\{\mathcal{C}_{rot}, \mathcal{C}_{id}\}$: the rotation angle for regression and the digit identity for classification.

Based on this, we evaluate two objective sets. Objective Set 1 uses the RNC loss for both concepts. Objective Set 2 uses RNC for the regression concept and SupCon for the classification concept. The complete results are presented in Table 4.

Table 3: Comprehensive comparison of representation quality, predictive performance, and concept composition on the MPIIGaze dataset.

| Objective Set | Method | Representation & Predictive Performance | | | | | | | | Concept Comp. |
| | | Spearman ↑ | | | V-score ↑ | MAE ↓ | | | Acc. ↑ | Sim. ↑ |
| | | x | y | z | id | x | y | z | id | ⊕ and ⊖ |
|---|---|---|---|---|---|---|---|---|---|---|
| | baseline | 0.54 | 0.48 | 0.53 | 0.18 | 0.01 | 0.02 | 0.02 | 0.49 | 0.14 |
| | FAMO | 0.52 | 0.40 | 0.52 | 0.23 | 0.01 | 0.03 | 0.03 | 0.18 | 0.09 |
| Objective Set 1 | Nash-MTL | 0.49 | 0.40 | 0.51 | 0.25 | 0.01 | 0.02 | 0.03 | 0.36 | 0.09 |
| | IMTL | 0.15 | 0.44 | 0.20 | 0.39 | 0.02 | 0.03 | 0.03 | 0.85 | 0.07 |
| | **Ours** | 0.72 | 0.92 | 0.79 | 0.99 | 0.01 | 0.02 | 0.02 | 0.99 | 0.95 |
| | baseline | 0.29 | 0.16 | 0.21 | 0.99 | 0.01 | 0.04 | 0.06 | 0.99 | 0.63 |
| | FAMO | 0.50 | 0.37 | 0.47 | 0.99 | 0.01 | 0.03 | 0.03 | 0.99 | 0.36 |
| Objective Set 2 | Nash-MTL | 0.36 | 0.45 | 0.32 | 0.99 | 0.01 | 0.03 | 0.03 | 0.99 | 0.29 |
| | IMTL | 0.28 | 0.51 | 0.29 | 0.99 | 0.01 | 0.03 | 0.03 | 0.99 | 0.35 |
| | **Ours** | 0.73 | 0.92 | 0.81 | 0.98 | 0.01 | 0.02 | 0.01 | 0.98 | 0.89 |

Table 4: Comprehensive comparison of representation quality, predictive performance, and concept composition on the Rotated MNIST dataset.

| Objective Set | Method | Representation & Predictive Performance | | | | Concept Comp. |
| | | Spearman ↑ | V-score ↑ | MAE (rad) ↓ | Acc. ↑ | Sim. ↑ |
| | | rot | id | rot | id | ⊕ and ⊖ |
|---|---|---|---|---|---|---|
| | baseline | 0.63 | 0.47 | 0.14 | 0.78 | 0.39 |
| | FAMO | 0.67 | 0.44 | 0.13 | 0.84 | 0.37 |
| Objective Set 1 | Nash-MTL | 0.62 | 0.42 | 0.14 | 0.82 | 0.38 |
| | IMTL | 0.57 | 0.47 | 0.14 | 0.87 | 0.39 |
| | **Ours** | 0.92 | 0.87 | 0.14 | 0.88 | 0.73 |
| | baseline | 0.02 | 0.96 | 0.73 | 0.99 | 0.71 |
| | FAMO | 0.59 | 0.96 | 0.14 | 0.99 | 0.54 |
| Objective Set 2 | Nash-MTL | 0.02 | 0.96 | 0.74 | 0.99 | 0.45 |
| | IMTL | 0.71 | 0.95 | 0.14 | 0.98 | 0.48 |
| | **Ours** | 0.93 | 0.96 | 0.12 | 0.99 | 0.71 |

## A.4 CONTINUAL LEARNING WITH DOMAIN EXPANSION

Our main paper assumes a fixed set of $M$ tasks defined *a priori*. A natural question is whether Domain Expansion can adapt to a continual learning setting where new tasks are added to a pre-trained model. We designed an experiment to validate this, demonstrating that our framework can successfully add $N$ new objectives $(C_{M+1}, ..., C_{M+N})$ to a model already pre-trained on $M$ tasks, without retraining from scratch.

Our continual learning approach is as follows:

1. **Freeze Existing Axes:** We "freeze" the original $M$ axes $(v_0...v_{M-1})$, which have already been trained.

2. **Find $N$ New Orthogonal Axes:** We find $N$ new axes by running eigendecomposition on the "residual" feature space $\mathcal{F}'$. This residual space is computed by removing the compo-

nents from the $M$ pre-trained axes:

$$f' = f - \sum_{i=0}^{M-1} f^{\text{proj},i}$$

We select the top $N$ eigenvectors from the covariance of $\mathcal{F}'$, which guarantees they are orthogonal to all previously learned axes.

3. **Train New Tasks (with Regularization):** We then train the $N$ new tasks. Each new objective $C_j$ (where $j \in \{M+1, ..., M+N\}$) is trained using its loss $\mathcal{L}_j$ on projections onto its corresponding new axis $v_j$.

4. **Prevent Catastrophic Forgetting:** To prevent the encoder from "forgetting" the original tasks, we add an L2 regularization loss. This loss ensures that the projected coefficients for the original $M$ tasks (on their frozen axes $v_0...v_{M-1}$) remain constant.

The total loss function for this new training phase, where $\mathcal{N}$ is the set of $N$ new tasks and $\mathcal{M}$ is the set of $M$ old tasks, is:

$$\mathcal{L}_{\text{total}} = \underbrace{\sum_{j \in \mathcal{N}} w_j \cdot \mathcal{L}_j(\mathcal{F}_j^{\text{proj}}, C_j)}_{\text{Loss for N new tasks}} + \lambda \underbrace{\sum_{m \in \mathcal{M}} \mathcal{L}_{\text{L2}}(\mathcal{E}_{\text{coeffs},m}, \bar{\mathcal{E}}_{\text{coeffs},m})}_{\text{Regularization for M old tasks}}. \quad (25)$$

Here, $\mathcal{L}_{\text{L2}}$ is the L2 (Mean Squared Error) loss, $\lambda$ is a balancing hyperparameter ($\lambda = 0.1$), $\mathcal{E}_{\text{coeffs},m}$ is the vector of coefficients for the $m$-th original task from the current encoder, and $\bar{\mathcal{E}}_{\text{coeffs},m}$ is the "frozen" target vector of coefficients captured before this new training began.

The results of this experiment are presented in Table 5. The "Continual" model, which was fine-tuned, shows a minor performance drop compared to the "Default" model trained on all tasks from scratch. This drop also affects the concept composition quality, which decreases from 0.95 to 0.72. However, the framework still successfully solves the latent collapse problem and prevents catastrophic forgetting. Crucially, even with this degradation, the model's performance on both representation (e.g., Spearman, V-score) and compositionality (0.72) remains remarkably stronger than the baselines in Table 1 (main paper), demonstrating the framework's viability for continual learning.

Table 5: Performance comparison of models trained from scratch versus continually. The "Continual" model was pre-trained on $\{az, el, cat\}$, then fine-tuned on $\{rot, id\}$. Arrows indicate whether higher (↑) or lower (↓) values are better.

| Objective Set | Method | Representation & Predictive Performance | | | | | | | | | | Concept Comp. |
| | | Spearman ↑ | | | V-score ↑ | | MAE° ↓ | | | Acc. ↑ | | Sim. ↑ |
| | | az | el | rot | cat | id | az | el | rot | cat | id | ⊕ and ⊖ |
| --- | --- | --- | --- | --- | --- | --- | --- | --- | --- | --- | --- | --- |
| Objective Set 1 | Default | 0.95 | 0.87 | 0.85 | 0.99 | 0.91 | 0.08 | 0.08 | 0.09 | 0.99 | 0.97 | 0.95 |
| | **Continual** | 0.92 | 0.81 | 0.84 | 0.82 | 0.93 | 0.11 | 0.11 | 0.12 | 0.94 | 0.87 | 0.72 |

## A.5 ANALYSIS OF EIGENVECTOR-TO-CONCEPT ASSIGNMENT

A key question regarding our framework is how each eigenvector $v_m$ is mapped to a specific task concept $C_m$. Our assignment is based on variance ranking (i.e., the top-$M$ eigenvectors are assigned to the $M$ objectives). However, the specific mapping (e.g., whether $v_0$ is assigned to *azimuth* or *category*) is arbitrary.

We hypothesize that this mapping is arbitrary because our training dynamic forces the alignment. The loss (as shown in Eq. 8 in the main paper) is computed after the projection. The gradient for a specific concept, $\mathcal{L}_m$, only backpropagates through its corresponding projected features, $f^{\text{proj},m}$, which lie on the axis $v_m$. This training dynamic forces the encoder to learn to map all variance related to concept $C_m$ onto whichever axis $v_m$ it has been assigned.

To prove this, we ran a new experiment where we **randomly** shuffled the eigenvector-to-concept assignments and retrained the model from scratch. We tested the default (variance-ranked) assignment

and two random shuffles (Default=$\{az, el, rot, cat, id\}$, Shuffle 1 = $\{cat, id, az, el, rot\}$, Shuffle 2 = $\{cat, id, az, el, rot\}$, assigned in order). The results are presented in Table 6.

Table 6: Comparison of model performance with the default variance-ranked assignment versus randomly shuffled assignments. All models are trained on Objective Set 1. Arrows indicate whether higher ($\uparrow$) or lower ($\downarrow$) values are better.

| Objective Set | Assignment | Representation & Predictive Performance | | | | | | | | | | Concept Comp. |
| --- | --- | --- | --- | --- | --- | --- | --- | --- | --- | --- | --- | --- |
| | | Spearman $\uparrow$ | | | V-score $\uparrow$ | | MAE° $\downarrow$ | | | Acc. $\uparrow$ | | Sim. $\uparrow$ |
| | | az | el | rot | cat | id | az | el | rot | cat | id | $\oplus$ and $\ominus$ |
| | Default | 0.95 | 0.87 | 0.85 | 0.99 | 0.91 | 0.08 | 0.08 | 0.09 | 0.99 | 0.97 | 0.95 |
| Objective Set 1 | **Shuffle 1** | 0.95 | 0.87 | 0.85 | 0.98 | 0.95 | 0.08 | 0.08 | 0.09 | 0.99 | 0.91 | 0.87 |
| | **Shuffle 2** | 0.95 | 0.87 | 0.85 | 0.98 | 0.94 | 0.08 | 0.08 | 0.09 | 0.99 | 0.91 | 0.88 |

As shown in the table, the final performance is not negatively impacted by the random shuffling. This confirms that the model is not relying on a pre-existing alignment of variance; rather, our method *creates* this alignment during training.

## A.6 Numerical Stability of Covariance Estimation

A critical component of our framework is the eigendecomposition of the latent feature covariance matrix (as described in the main paper, Equation 3). The stability of this estimation is crucial for a stable orthogonal basis.

A naive approach would be to estimate the covariance matrix on each training mini-batch (e.g., $B_{\text{train}} = 256$). However, this introduces significant instability. Our latent dimension is $D = 2048$, and a typical training batch is much smaller than the dimensionality ($B_{\text{train}} \ll D$). This is a classic "small $n$, large $p$" problem, where estimating a $D \times D$ covariance matrix from $n$ samples is statistically ill-posed. The resulting matrix is noisy and singular, and its eigenvectors are unstable, preventing the model from converging.

To solve this, we decouple the training batch size ($B_{\text{train}}$) from the covariance estimation set size ($B_{\text{cov}}$).

Our final methodology is as follows:

1. At the beginning of each training epoch, we first compute latent features $\mathcal{F}$ for a large, stable subset of the training data, defined by $B_{\text{cov}}$.

2. We compute the covariance matrix and its eigenvectors $V_M$ on this large, stable batch $\mathcal{F}$.

3. For all training mini-batches (of size $B_{\text{train}}$) within that epoch, we **freeze** this orthogonal basis $V_M$.

To determine how large $B_{\text{cov}}$ must be, we ran an ablation study varying its size as a percentage of the total training set. The results are shown in Table 7. The performance is remarkably stable. Even when using only 10% of the training data for the covariance estimation, the performance remains high. This confirms that our method is not only numerically stable, as a stable basis can be estimated from a small subset of the data.

## A.7 Robustness to Correlated or Redundant Tasks

A crucial question is whether our framework's strict orthogonality assumption is suboptimal or even detrimental when tasks are highly correlated or redundant. The concern is that enforcing orthogonality may "cost" the model, hurting performance on related tasks that could otherwise benefit from a shared representation.

To test this, we designed a new experiment where we explicitly introduced task redundancy. We took all 5 objectives from Objective Set 1 (az, el, rot, cat, id) and duplicated them, creating a new set of 5 "copy" tasks.

Table 7: Ablation study on the effect of the covariance estimation set size ($B_{cov}$), measured as a percentage of the total training set. All models are trained on Objective Set 1. Arrows indicate whether higher (↑) or lower (↓) values are better.

| Objective Set | $B_{cov}$ (% of Train) | Representation & Predictive Performance | | | | | | | | | | Concept Comp. |
|---|---|---|---|---|---|---|---|---|---|---|---|---|
| | | Spearman ↑ | | | V-score ↑ | | MAE° ↓ | | | Acc. ↑ | | Sim. ↑ |
| | | az | el | rot | cat | id | az | el | rot | cat | id | ⊕ and ⊖ |
| Objective Set 1 | 10% | 0.96 | 0.87 | 0.85 | 0.99 | 0.94 | 0.08 | 0.08 | 0.09 | 0.99 | 0.97 | 0.94 |
| | 20% | 0.96 | 0.87 | 0.87 | 0.98 | 0.95 | 0.08 | 0.08 | 0.09 | 0.99 | 0.97 | 0.95 |
| | 50% | 0.95 | 0.87 | 0.85 | 0.98 | 0.93 | 0.08 | 0.08 | 0.09 | 0.98 | 0.95 | 0.97 |
| | 100% (Default) | 0.95 | 0.87 | 0.85 | 0.99 | 0.91 | 0.08 | 0.08 | 0.09 | 0.99 | 0.97 | 0.95 |

We then trained a new model on all 10 tasks simultaneously, assigning each of the 10 tasks to its own unique, orthogonal axis.

The results are presented in Table 8. We compare the performance of the default model (trained on 5 tasks) against the new model (trained on 10 tasks). The key finding is that the performance on the **original 5 tasks** remains stable, with no negative impact from the addition of the 5 redundant tasks. This suggests that our framework is robust to task redundancy and that the "cost" of enforcing orthogonality, even for highly correlated tasks, is not detrimental.

Table 8: Robustness to redundant tasks. We compare the default 5-task model against a model trained on 10 tasks (the 5 original + 5 duplicates). Performance on the original 5 tasks is not impacted, and the model successfully learns the new redundant tasks. Arrows indicate whether higher (↑) or lower (↓) values are better. (Note: The composition of the redundant task is reconstructed by 10 eigenvectors.)

| Objective Set | Method | Representation & Predictive Performance | | | | | | | | | | Concept Comp. |
|---|---|---|---|---|---|---|---|---|---|---|---|---|
| | | Spearman ↑ | | | V-score ↑ | | MAE° ↓ | | | Acc. ↑ | | Sim. ↑ |
| | | az | el | rot | cat | id | az | el | rot | cat | id | ⊕ and ⊖ |
| Objective Set 1 | Default | 0.95 | 0.87 | 0.85 | 0.99 | 0.91 | 0.08 | 0.08 | 0.09 | 0.99 | 0.97 | 0.95 |
| | **Original 5** | 0.94 | 0.85 | 0.84 | 0.98 | 0.83 | 0.09 | 0.09 | 0.10 | 0.99 | 0.99 | 0.91 |
| | **Redundant 5** | 0.85 | 0.84 | 0.84 | 0.98 | 0.83 | 0.09 | 0.09 | 0.10 | 0.97 | 0.94 | 0.91 |