# OpenReview forum: "Domain Expansion: A Latent Space Construction Framework for Multi-Task Learning"
_ICLR.cc/2026/Conference — ICLR 2026 Poster_

### Official Review · Reviewer_wQsh · 2025-10-17

**Soundness:** 2
**Presentation:** 3
**Contribution:** 2
**Rating:** 2
**Confidence:** 4

**Summary:**

This paper introduces "Domain Expansion," a novel framework for multi-task learning (MTL) designed to address what the authors term "latent representation collapse." This phenomenon occurs when a single network trained on multiple objectives learns a compromised and suboptimal shared representation due to conflicting gradients. The core idea of Domain Expansion is to prevent this interference structurally by constructing a latent space where each learning objective is assigned to a dedicated, mutually orthogonal subspace. This is achieved through a three-step process at each training epoch: (1) finding the principal axes of the latent feature distribution via eigendecomposition of the covariance matrix, (2) defining an orthogonal domain by assigning the top eigenvectors to different tasks, and (3) using "orthogonal pooling" to project the shared latent feature into these concept-specific subspaces before decoding.

The authors validate their approach on several benchmarks (ShapeNet, MPIIGaze, Rotated MNIST) that combine classification and regression tasks. The experiments show that Domain Expansion not only outperforms standard MTL baselines and gradient-based conflict mitigation methods in both representation quality and predictive performance but also creates a structured and compositional latent space. This structured space allows for algebraic operations on concepts, which the authors demonstrate through concept composition experiments.

**Strengths:**

* The concept of "latent representation collapse" is well-motivated and clearly illustrated in Figure 1. The paper does an excellent job of formalizing this problem and positioning Domain Expansion as a direct solution.
* A key strength of the proposed method is its ability to create a latent space that is not a "black box." The paper demonstrates that the resulting orthogonal structure enables meaningful, algebraic manipulation of concepts (e.g., concept-specific adjustment and composition), a significant step towards more controllable and interpretable models.

**Weaknesses:**

*   **Originality and Relation to Prior Work:** The core idea of representing concepts in a latent space may not be entirely new. The paper could be strengthened by discussing its relationship to existing concepts such as superposition in neural networks[1,2], embedding properties in word vectors [3], and the linear representation hypothesis [4].
*   **Limited Scope of Empirical Validation:** The experiments, while well-executed, are confined to controlled vision datasets (ShapeNet, Rotated MNIST, MPIIGaze). The paper makes broad claims about applicability, but there is no evaluation on other domains like text or multi-modal learning, or on large-scale, complex benchmarks. Furthermore, the lack of an ablation study or resource analysis makes it difficult to assess the scalability and efficiency of the eigendecomposition and Hungarian alignment steps, limiting the claims of general applicability.
*   **Insufficient Theoretical Grounding for Orthogonal Basis Stability:** The framework relies on a dynamic eigendecomposition performed at each epoch. This raises concerns about the stability of the basis vectors, which could shift and introduce noise into the training process.
*   **Missing Discussion on Computational and Memory Overheads:** The proposed method introduces a covariance matrix calculation and eigendecomposition step (Equations 3-4) in each training epoch. For high-dimensional latent spaces (D=2048 in this paper), this introduces a significant computational complexity of O(D³), which is not analyzed or compared against the runtime costs of gradient-based MTL methods. This omission may hinder the method's reproducibility and adoption for larger models where this overhead could be prohibitive.
*   **Minor Ambiguities in Mathematical Formulation:** While the mathematical exposition is generally clear, there are some inconsistencies in notation. For instance, the variable 'F' is used to denote the latent space, a batch of representations, and a single latent vector at different points. Similarly, the formulation of the total loss in Equation (8) seems inconsistent with the definition of the projected subspace in Equation (5), as the loss is computed on projected vectors, not the subspace itself. The notation for the set of objectives 'M' is also used ambiguously. These minor issues could be tightened for improved clarity.

[1] Toy Models of Superposition

[2] Linear algebraic structure of word senses, with applications to polysemy

[3] Linguistic regularities in continuous space word representations

[4] The Linear Representation Hypothesis and the Geometry of Large Language Models

**Questions:**

See Weaknesses

---

> ### Author Response · Authors · 2025-11-21
> **Respond to Reviewer wQsh (Part 1)**
>
> We sincerely thank Reviewer wQsh for their thorough and challenging critique. We are encouraged that you recognized the core problem of "latent representation collapse" as **"well-motivated"** and our method for creating a non-"black box" space as a **"significant step"** toward interpretability. Your concerns about originality, cost, and stability are all valid and critical. We have new experimental results (in the **supplementary material**) and have clarified existing sections to address every point.
>
> **Q1. Originality and Relation to Prior Work: The core idea of representing concepts in a latent space may not be entirely new. The paper could be strengthened by discussing its relationship to existing concepts such as superposition in neural networks[1,2], embedding properties in word vectors [3], and the linear representation hypothesis [4].**
>
> **Ans1:** We thank you for these insightful references, which we will *add to our related work*. You are right that the ideas of linear structure are related. However, we wish to clarify a fundamental distinction:
>
> Works like [3, 4] (word vectors, linear rep. hypothesis) are primarily **observational**: they find that linear structures emerge in trained models. Our work is **constructive** and **prescriptive**. Domain Expansion is an architectural framework that *proactively* constructs and enforces an orthogonal, compositional structure by design. We do not hope this structure emerges; we *build* it explicitly to solve the specific problem of task interference in **multi-task learning**.
>
> Furthermore, those works observe structure in a single, unsupervised domain (language), whereas our framework is designed to impose this structure to solve multi-task conflicts across supervised objectives (including both **regression** and **classification**). This constructive mechanism for **preventing representation collapse** is the core novelty.
>
> **Q2. Limited Scope of Empirical Validation: The experiments, while well-executed, are confined to controlled vision datasets (ShapeNet, Rotated MNIST, MPIIGaze). The paper makes broad claims about applicability, but there is no evaluation on other domains like text or multi-modal learning, or on large-scale, complex benchmarks. Furthermore, the lack of an ablation study or resource analysis makes it difficult to assess the scalability and efficiency of the eigendecomposition and Hungarian alignment steps, limiting the claims of general applicability.**
>
> **Ans2:** We thank you for this multi-part critique. You are correct that the original submission was missing key ablations on stability and efficiency. We have run these new experiments and now present them, along with our reasoning for the dataset scope.
>
> **1. Stability & Efficiency Analyses (New Experiments):**
>
> *a. Eigendecomposition Stability:* We ran a new ablation study (**Table 7, Appendix A.6**) on the size of the dataset used for covariance estimation ($B_{\text{cov}}$). We found the method is remarkably stable and efficient, with performance remaining high even when using only 10% of the training data. At 10%, **Spearman scores are $\approx$ 0.9** or higher, **V-scores are $>$ 0.95**, **MAE is $\approx$ 0.08**, and **Accuracy is $>$ 0.95**, all nearly identical to using 100% of the data.
>
> *b. Hungarian Alignment Stability:* To evaluate the stability of the basis vectors, we computed the cosine similarity of the basis between consecutive epochs (**Fig. 8, Appendix A.2**). The analysis shows the basis is adaptive in early training but stably converges, **approaching 1.0 similarity around epoch 60 and reaching 0.99 by epoch 100**. These new results confirm the scalability and efficiency of our approach.
>
> **2. Dataset Scope (Why we chose these datasets):**
>
> *a. On Text/Multi-modal Domains:* Our paper's scope, as stated in the abstract and introduction, is focused on solving representation collapse within a single, unified multi-task network. While we believe the principles could apply elsewhere, text and multi-modal learning (which have their own specific challenges) were outside the scope of this foundational work.
>
> *b. On Large-Scale Benchmarks:* We agree that scaling to complex benchmarks is a critical next step. For this paper, we intentionally chose datasets (ShapeNet, MPIIGaze) that provided **clean, verifiable conceptual axes** (e.g., pose, gaze). This controlled setting was essential to rigorously prove our paper's two core, novel claims: first, that we can **verifiably prevent latent space collapse**, and second, that our method creates a **functional, compositional algebra**.This is a precise claim that would be very difficult to measure and verify in a more complex, entangled dataset, and we felt it was necessary to prove the concept in this clear setting first. We will highlight scaling as the primary direction for future work.

---

> ### Author Response · Authors · 2025-11-21
> **Respond to Reviewer wQsh (Part 2)**
>
> **Q3. Insufficient Theoretical Grounding for Orthogonal Basis Stability: The framework relies on a dynamic eigendecomposition performed at each epoch. This raises concerns about the stability of the basis vectors, which could shift and introduce noise into the training process.**
>
> **Ans3:** This is a crucial concern. As detailed in our answer to Q2 and shown in our new stability analysis in **Appendix A.2 (Fig. 8)**, we analyzed this exact problem. We found that while the basis is dynamic and adaptive in the early phases of training (as the encoder learns), it stably converges as training progresses. It **approaches 1.0 cosine similarity around epoch 60 and is 0.99 by epoch 100**, confirming the basis and the encoder's representation successfully stabilize.
>
> **Q4. Missing Discussion on Computational and Memory Overheads: The proposed method introduces a covariance matrix calculation and eigendecomposition step (Equations 3-4) in each training epoch. For high-dimensional latent spaces (D=2048 in this paper), this introduces a significant computational complexity of O(D³), which is not analyzed or compared against the runtime costs of gradient-based MTL methods. This omission may hinder the method's reproducibility and adoption for larger models where this overhead could be prohibitive.**
>
> **Ans4:** Thank you for raising this important practical concern. We analyze this cost in two parts:
> 1. **Training**: You are correct that eigendecomposition adds overhead. To precisely quantify this, we ran a new analysis comparing the training time and memory of our method against the baseline. Our experiments were run on a server with an AMD EPYC 7413 24-Core Processor and an NVIDIA A100 GPU. We trained the encoder on Objective Set 1 for 100 epochs with a batch size of 256. The table below shows the average results per epoch. As the results show, our method adds only a manageable computational overhead (approx. 20% increase in epoch time) while having a negligible impact on peak GPU memory. This cost is due to the $O(D^3)$ eigendecomposition on the 2048-D features. **We also note that this eigendecomposition is computed on the CPU, so this 20% time overhead does not increase GPU load**.
>
> | Method | Avg. Epoch Time | Max GPU Memory |
> | :--- | :---: | :---: |
> | Baseline | 1.95 min | 11.2 GB |
> | Ours (Domain Expansion) | 2.34 min | 11.3 GB |
>
> 2. **Inference**: This is a key strength of our approach. After training, the orthogonal basis $V_M$​ is **fixed**. An inference pass only requires the standard encoder forward pass and a simple, computationally lightweight projection. Thus, **inference speed is not affected** and is virtually identical to the baseline.
>
> We believe this manageable, one-time training cost is a reasonable trade-off for the significant gains in interpretability and compositionality.
>
> **Q5: Minor Ambiguities in Mathematical Formulation: While the mathematical exposition is generally clear, there are some inconsistencies in notation. For instance, the variable 'F' is used to denote the latent space, a batch of representations, and a single latent vector at different points. Similarly, the formulation of the total loss in Equation (8) seems inconsistent with the definition of the projected subspace in Equation (5), as the loss is computed on projected vectors, not the subspace itself. The notation for the set of objectives 'M' is also used ambiguously. These minor issues could be tightened for improved clarity.**
>
> **Ans:** Thank you for this detailed feedback. You are absolutely correct that our notation should be more precise, and we apologize for these ambiguities. This is an oversight on our part.
> In the final camera-ready version, we will meticulously revise the mathematical formulation to ensure: (1) $f$ is reserved for a single vector and $\mathcal{F}$ is used for a set of representations, (2) the loss in Eq. (8) is clearly written as a function of the projected vectors ($f^{\text{proj,m}}$), and (3) $M$ is unambiguously defined. We appreciate you catching these details.

---

> > ### Comment · Reviewer_wQsh · 2025-11-26
> >
> > Thank you for your response. While some of my initial concerns have been addressed, several critical issues remain unclear:
> >
> > 1. Relegating complex tasks to future work significantly weakens the validation of the method. Without testing on more challenging benchmarks, it is unclear if this approach provides a genuine improvement over standard baselines in practical settings.
> > 2. The introduction claims that competing objectives degrade representations by pulling them in opposing directions. This contradicts the empirical success of Foundation Models, where multi-objective training often yields robust, transferable features. Regarding the specific failure modes in pose/gaze estimation: have the authors ruled out confounding factors such as low data diversity, improper weight initialization, or insufficient latent dimensionality? The collapse shown in the baselines could easily be attributed to these factors rather than fundamental gradient conflicts. Furthermore, the visualization in Figure 2 is an oversimplification that does not necessarily hold in high-dimensional spaces.
> > 3. The paper argues that explicit orthogonal construction is necessary to prevent interference. However, literature such as “Toy Models of Superposition” (which I refer to) and the Johnson-Lindenstrauss lemma indicate that high-dimensional vectors are already likely to be approximately orthogonal. Since LLMs and other large models successfully encode vast amounts of information without this specific architectural constraint, the justification for why the models in this paper collapse seems insufficient.

---

> ### Author Response · Authors · 2025-11-27
> **Respond to Reviewer wQsh (Part 3)**
>
> **Added Q1: Relegating complex tasks to future work significantly weakens the validation of the method. Without testing on more challenging benchmarks, it is unclear if this approach provides a genuine improvement over standard baselines in practical settings.**
>
> **Ans 1:** We thank the reviewer for this crucial point regarding our validation strategy. We agree that scaling to complex benchmarks is an important future direction.
> However, for this work, our choice of datasets (ShapeNet, MPIIGaze) was a **deliberate and necessary scientific choice, not a compromise.**
>
> **Our primary requirement was to "clearly show and solve the problem."** These datasets provide **clean, verifiable conceptual axes** (e.g., pose, gaze), making them far more suitable for rigorously validating our core claims. This controlled setting was the only way to verifiably prove our paper's two novel contributions:
> 1. That we can prevent the **latent space collapse**.
> 2. That our method creates a **functional, compositional algebra.**
>
> These are precise claims that would be difficult to measure and verify in a more complex, entangled dataset. Therefore, we felt it was scientifically necessary to first prove that the mechanism works and why it works in this clear, falsifiable setting. Addressing the verification challenges in these more complex settings is a key part of our plan for future work.

---

> ### Author Response · Authors · 2025-11-27
> **Respond to Reviewer wQsh (Part 4)**
>
> **Added Q2: The introduction claims that competing objectives degrade representations by pulling them in opposing directions. This contradicts the empirical success of Foundation Models, where multi-objective training often yields robust, transferable features. Regarding the specific failure modes in pose/gaze estimation: have the authors ruled out confounding factors such as low data diversity, improper weight initialization, or insufficient latent dimensionality? The collapse shown in the baselines could easily be attributed to these factors rather than fundamental gradient conflicts. Furthermore, the visualization in Figure 2 is an oversimplification that does not necessarily hold in high-dimensional spaces.**
>
> **Ans2:** This is a critical, multi-part question. To provide the clearest answer, we first want to **precisely define the "latent representation collapse"** we are solving, as it is different from the generalist training of a Foundation Model.
> Our work addresses conflicts between heterogeneous, structured losses (defined in eq. 2). For example, our baseline uses Rank-N-Contrast (RNC) loss, which **proactively** forces the latent space to rank features according to a specific label. When multiple RNC losses are active (e.g., Task 1 ranks by 'pose', Task 2 ranks by 'gaze'), their gradients are in direct conflict. The features cannot simultaneously satisfy both ranking requirements, leading to a suboptimal, entangled space. **This failure to achieve the required proactive structure is what we define as ''latent representation collapse.''**
>
> This is a fundamentally different challenge than the one solved by most Foundation Models. Their goal is often general, *robust feature extraction* from (often-aligned) objectives. Our goal is *explicit, controllable, multi-axis separation* for heterogeneous, structured tasks. The "collapse" we solve is a failure to impose this specific, proactive structure.
> This is why our core metrics are **Spearman's $\rho$ (for ranking) and V-score (for binarizing)**. As shown in Table 1, the baselines fail on these metrics, which quantitatively demonstrates the collapse we have defined.
>
> With this precise definition, we can now address the reviewer's specific points:
>
> 1. **On Confounding Factors (Tuning, Data, Dim):** We rigorously controlled for these factors. All methods (ours and baselines) use the same standard ResNet-50 backbone and the same 2048 latent dimension. Crucially, as detailed in Sec.4, we **carefully tuned all loss weights $w_m$ to ensure the magnitude of each task's loss was balanced.** For example, the SupCon loss (scale $\approx$200.0) was weighted by 0.02 to balance the RNC loss (scale $\approx$5.0). This was done for all methods. The baselines still collapsed. This strongly suggests the collapse is a fundamental problem of competing proactive objectives, not an artifact of poor tuning.
>
> 2. **On Foundation Models:** As established above, our paper addresses the specific challenge of heterogeneous MTL (e.g., ranking + classification), where objectives are not aligned and may be fundamentally in conflict. Our experiments (and the large body of work on gradient-based MTL like PCGrad, FAMO, etc.) confirm that in this specific, common setting, representation collapse is a very real problem.
>
> 3. **On Figure 2:** The reviewer is correct that Figure 2 is a 2D projection. It is not meant as high-dimensional proof, but as a visualization of the quantitative collapse shown in our tables (e.g., the low Spearman's $\rho$ and V-Score of the baselines). It illustrates that the baseline features, when projected, are entangled, while ours remain separable.

---

> ### Author Response · Authors · 2025-11-27
> **Respond to Reviewer wQsh (Part 5)**
>
> **Added Q3: The paper argues that explicit orthogonal construction is necessary to prevent interference. However, literature such as “Toy Models of Superposition” (which I refer to) and the Johnson-Lindenstrauss lemma indicate that high-dimensional vectors are already likely to be approximately orthogonal. Since LLMs and other large models successfully encode vast amounts of information without this specific architectural constraint, the justification for why the models in this paper collapse seems insufficient.**
>
> Ans3: This is a fascinating theoretical point, and we thank the reviewer for raising it. We believe this distinction is the single most important one for understanding our contribution.
>
> The reviewer is correct that high-dimensional spaces are statistically "likely" to contain many "approximately" orthogonal vectors. However, as we established in our answer to Q2, this is a **passive, statistical property** of the space.
>
> The problem we solve is **proactive**, not passive.
>
> As we defined, our baselines use losses like SupCon and RNC, which are **actively trying to impose a specific, non-random structure** on the latent space (e.g., "rank all features by pose").
>
> The "passive" approximate orthogonality of the space is irrelevant to this challenge. These **random, semantically-meaningless** vectors cannot align with the specific, non-random structure the RNC loss is trying to create.
>
> When two proactive losses (like RNC for 'pose' and RNC for 'gaze') try to impose *different* structured alignments on the same space, they conflict and collapse. The "passive" background orthogonality of the space does nothing to stop this conflict.
>
> Our method solves this **proactive-vs-proactive** conflict. It is also a **proactive construction.** Instead of letting the losses fight, our method proactively builds a **specific, interpretable basis** and gives each proactive loss (RNC, SupCon) its own exclusive, orthogonal axis to control.
>
> This explicit construction is what guarantees that the 'pose' axis is not entangled with the 'gaze' axis. It's how we verifiably prevent the specific collapse we defined (Eq. 2), which is something the 'passive' properties of the space cannot do.
> Furthermore, this proactive construction is what enables the specific properties of our framework, such as **supporting both continuous (regression) and binary (classification) tasks on different axes, which a passive, random basis cannot provide.**

---

### Official Review · Reviewer_CqRr · 2025-10-28

**Soundness:** 2
**Presentation:** 3
**Contribution:** 2
**Rating:** 4
**Confidence:** 4

**Summary:**

The paper introduces Domain Expansion, a novel framework designed to address the problem of "latent representation collapse" in multi-objective learning. The core problem is that when a single network is trained on multiple tasks, conflicting gradients can lead to a compromised latent space that is suboptimal for all tasks. The authors' propose to structure the latent space into a set of mutually orthogonal subspaces, one for each objective. By using an orthogonal pooling mechanism, the method ensures that the learning signals for each task are decoupled, preventing interference by design. The paper demonstrates that this approach not only improves performance on multi-objective benchmarks but also creates an interpretable and compositional latent space where concepts can be manipulated algebraically.

**Strengths:**

* **S1. Clarity and Presentation:** The paper is well-written and easy to follow. The authors do a great job of motivating the problem with the clear concept of "latent representation collapse." The flow from the problem statement to the proposed method is logical and intuitive. The figures are highly effective at illustrating the core idea.
* **2. Originality and Elegance of the Method:** The central idea of harnessing the principal eigenvectors of the latent space's covariance matrix to form an orthogonal basis for different tasks is, to my knowledge, novel. It is a novel approach that addresses the root cause of task interference rather than reactively managing gradient conflicts during optimization, which is the focus of many existing methods.
* **3. Interpretability and Compositionality:** A significant strength of this work is that it doesn't just aim for better metrics, but for a more structured and meaningful latent space. The demonstration of a "concept algebra", where latent vectors can be manipulated through simple arithmetic to compose or adjust concepts, is a contribution towards more interpretable and controllable models.

**Weaknesses:**

* **W1. Scope of Empirical Evaluation:** The experimental validation, while thorough on the chosen dataset, could be broadened to better establish the generalizability of the method.
    * **Architectural Diversity**: The experiments are conducted using only a ResNet-50 encoder. It would be beneficial to understand if the method is equally effective with other modern architectures, such as Vision Transformers, which may have different latent space geometries.
    * **Dataset Diversity**: The primary results are demonstrated on ShapeNet. While MPIIGaze and Rotated MNIST are mentioned in the appendix, a more in-depth analysis on datasets with different characteristics (e.g., more complex scenes, different data modalities) would strengthen the paper's claims.
    * **Robustness of Results**: The results in Table 1 are reported without standard deviations across multiple runs. Including these would provide greater confidence in the stability and significance of the performance gains.
* **W2. Clarity of Results in Table 1:** The main results table is dense and could be made more reader-friendly.
    * **Readability**: A suggestion would be to add a horizontal line to visually separate the baseline methods from the proposed method. Highlighting the best-performing score in each column (e.g., in bold) would also allow for a quicker assessment of the results.
    * **Caption Details**: The caption could be improved by explicitly defining the acronyms used in the column headers (e.g., az, el, rot, cat, id), as they are not defined in the main text of the experiments section.
* **W3. Nuanced Performance on Objective Set 2:** The results for Objective Set 2 are interesting and warrant more discussion. While the proposed method clearly excels on representation quality metrics (Spearman, V-score), some baselines achieve slightly better or comparable predictive performance on the downstream tasks (e.g., MAE, Accuracy). This raises the question of the practical trade-offs involved and in which scenarios superior representation quality is most critical.
* **W4. Reproducibility:** The paper does not mention the release of code. Providing the implementation would be a valuable contribution to the community, allowing others to build upon this work.

**Questions:**

* **Q1. Generalizability to Other Architectures:** Could the authors comment on the applicability of Domain Expansion to other encoder architectures, such as Transformers? Does the core assumption that principal eigenvectors correspond to meaningful, disentanglable concepts, hold in those architectures as well?

* **Q2. Performance on Standard Multi-Task Benchmarks:** To better validate the proposed method, it would be insightful to test the model on more complex, standard multi-task benchmarks like NYUv2, CelebA, or Cityscapes. Would the authors anticipate any specific challenges, for instance, with a much larger number of tasks (as in CelebA) or with dense prediction tasks (like semantic segmentation in Cityscapes)?

* **Q3. Transferability of the Learned Representation:** Could a model pre-trained with Domain Expansion be adapted to new tasks or datasets? For example, could a model trained on concepts {azimuth, elevation, category} be effectively fine-tuned on a new dataset that only has labels for {azimuth, category}, or perhaps one where a new concept, {color}, is introduced?

* **Q4. Training Dynamics and Computational Cost:** How many epochs are typically needed for the orthogonal basis to stabilize before the encoder is frozen? What's the time per epoch? What is the computational overhead of the eigendecomposition step per epoch compared to a standard multi-task baseline?

* **Q5. The Value of Representation Quality:** The results for Objective Set 2 compellingly show that baseline methods can achieve high predictive accuracy despite having a collapsed latent representation (i.e., poor V-scores). However, in a practical scenario where a user only cares about the final prediction accuracy on this specific test set, what is the key argument for your method? Does the superior representation quality translate to other critical benefits, such as improved robustness, better generalization to out-of-distribution samples, or greater fairness?

---

> ### Author Response · Authors · 2025-11-21
> **Response to Reviewer CqRr (Part 1)**
>
> We sincerely thank Reviewer CqRr for their thoughtful and constructive review. We are delighted that you found the paper **"well-written,"** the problem of "latent representation collapse" **"clear,"** and the core idea **"novel"** and **"elegant."** Your feedback is focused on generalizability and the practical value of a good representation, which are the most critical points. We have new experimental results (in the **supplementary material**) that directly address these concerns.
>
> **Q1. Generalizability to Other Architectures: Could the authors comment on the applicability of Domain Expansion to other encoder architectures, such as Transformers? Does the core assumption that principal eigenvectors correspond to meaningful, disentanglable concepts, hold in those architectures as well?**
>
> **Ans1:** This is a key question. To answer it, we first ran a new baseline experiment to see if the objective functions we use (SupCon, RNC) are even compatible with a ViT backbone. We found that running a standard ViT baseline with just the SupCon or RNC losses (i.e., without our Domain Expansion framework) does not converge.
>
> We are also actively investigating the cause, though we do not have a definitive solution yet. Our initial investigation into the literature suggests this may be a non-trivial and broader challenge. For example, we noted that the original papers for our objective functions, SupCon (Khosla et al., 2020) and Rank-N-Contrast (Zha et al., 2023), do not report results on transformer-based backbones.
>
> We hypothesize that these losses, which were designed for the spatial, global average-pooled features of a CNN, may not be "plug-and-play" compatible with the token representation of a ViT. token architecture.
> Therefore, adapting these specific losses to transformer architectures appears to be an independent and open research problem, which we will highlight as a direction for future work in our discussion.
>
> **Q2. Performance on Standard Multi-Task Benchmarks: To better validate the proposed method, it would be insightful to test the model on more complex, standard multi-task benchmarks like NYUv2, CelebA, or Cityscapes. Would the authors anticipate any specific challenges, for instance, with a much larger number of tasks (as in CelebA) or with dense prediction tasks (like semantic segmentation in Cityscapes)?**
>
> **Ans2:** We thank the reviewer for this important point and agree that scaling to more complex benchmarks is a critical next step. For this foundational work, we intentionally chose datasets (ShapeNet, MPIIGaze) that provided **clean, verifiable conceptual axes** (e.s., pose, gaze) . This controlled setting was essential to rigorously demonstrate our paper's core claims: first, that we can **verifiably prevent collapse**, and second, that we can build a functional, **compositional algebra**, which would be difficult to measure in a more entangled dataset like CityScapes.
>
> We believe this paper provides the necessary proof-of-concept, and we will be sure to highlight scaling to these more complex benchmarks as the primary direction for future work in our discussion.

---

> ### Author Response · Authors · 2025-11-21
> **Response to Reviewer CqRr (Part 2)**
>
> **Q3. Transferability of the Learned Representation: Could a model pre-trained with Domain Expansion be adapted to new tasks or datasets? For example, could a model trained on concepts {azimuth, elevation, category} be effectively fine-tuned on a new dataset that only has labels for {azimuth, category}, or perhaps one where a new concept, {color}, is introduced?**
>
> **Ans3:** This is an excellent question about our framework's flexibility. While our paper's main experiments assume a fixed set of M tasks, we designed a new experiment specifically for this rebuttal to prove that Domain Expansion can be adapted to a continual learning setting. We successfully added N new objectives ($C_{M+1},...,C_{M+N}$​) to a model already pre-trained on M tasks, without retraining from scratch.
>
> Our approach is as follows:
>
> 1. *Freeze Existing Axes:* We "freeze" the original $M$ axes ($v_0​...v_{M−1}$​) and initialize the encoder, which have already been trained.
>
> 2. *Find N New Orthogonal Axes:* We find $N$ new axes by running eigendecomposition on the "residual" feature space $F’$ (where $f^′=f−\sum_{i=0}^{M-1} f^{proj,i}$). We select the top N eigenvectors from this residual space, which ensures they are orthogonal to all previously learned axes.
>
> 3. *Train New Tasks (with Regularization):* We then train the $N$ new tasks. Each new objective $C_{M+j}$​ is trained using its loss $L_{M+j}$​ on projections onto its corresponding new axis $v_{M+j−1}$​
>
> 4. *Prevent Catastrophic Forgetting:* To prevent the encoder from "forgetting" the original tasks (i.e., catastrophic forgetting), we add an L2 regularization loss. This loss ensures that the projected coefficients for the original $M$ tasks (on their frozen axes $v_0​...v_{M−1}$​) remain constant while the $N$ new tasks are being learned.
>
> The total loss function for this new training phase is defined **eq.(25) Appendix A.4**.
>
> The results of this experiment are presented in **Table 5, Appendix A.4**. The "Continual" model, which was fine-tuned, shows a **minor** performance drop compared to the "Default" model trained on all tasks from scratch. This drop also affects the concept composition quality, which decreases from 0.95 to 0.72. However, the framework still successfully **solves the latent collapse problem** and **prevents catastrophic forgetting**. Crucially, even with this degradation, the model's performance on both representation (e.g., **Spearman, V-score > 0.8**) and compositionality (**0.72**) remains remarkably stronger than the baselines in Table 1 (main paper), demonstrating the framework's viability for continual learning.
>
> **Q4. Training Dynamics and Computational Cost: How many epochs are typically needed for the orthogonal basis to stabilize before the encoder is frozen? What's the time per epoch? What is the computational overhead of the eigendecomposition step per epoch compared to a standard multi-task baseline?**
>
> **Ans4:** Thank you for raising this important practical concern. First, To evaluate training dynamics and Hungarian alignment, we compute the cosine similarity of the basis between consecutive epochs. As shown in **Fig. 8 (Appendix A.2)**, the basis is adaptive during the early phases of training, but it **stably converges, approaching 1.0 similarity around epoch 60**. By epoch 100, the similarity is **0.99** (shown in Table 2), which means the orthogonal basis and the encoder's representation have successfully stabilized.
>
> Second, we analyze the computing cost in two parts:
>
> 1. **Training**: You are correct that eigendecomposition adds overhead. To precisely quantify this, we ran a new analysis comparing the training time and memory of our method against the baseline. Our experiments were run on a server with an AMD EPYC 7413 24-Core Processor and an NVIDIA A100 GPU. We trained the encoder on Objective Set 1 for 100 epochs with a batch size of 256. The table below shows the average results per epoch. As the results show, our method adds only a manageable computational overhead (approx. 20% increase in epoch time) while having a negligible impact on peak GPU memory. This cost is due to the $O(D^3)$ eigendecomposition on the 2048-D features. **We also note that this eigendecomposition is computed on the CPU, so this 20% time overhead does not increase GPU load**.
>
> | Method | Avg. Epoch Time | Max GPU Memory |
> | :--- | :---: | :---: |
> | Baseline | 1.95 min | 11.2 GB |
> | Ours (Domain Expansion) | 2.34 min | 11.3 GB |
>
> 2. **Inference**: This is a key strength of our approach. After training, the orthogonal basis $V_M$​ is **fixed**. An inference pass only requires the standard encoder forward pass and a simple, computationally lightweight projection. Thus, **inference speed is not affected** and is virtually identical to the baseline.
>
> We believe this manageable, one-time training cost is a reasonable trade-off for the significant gains in interpretability and compositionality.

---

> ### Author Response · Authors · 2025-11-21
> **Response to Reviewer CqRr (Part 3)**
>
> **Q5. The Value of Representation Quality: The results for Objective Set 2 compellingly show that baseline methods can achieve high predictive accuracy despite having a collapsed latent representation (i.e., poor V-scores). However, in a practical scenario where a user only cares about the final prediction accuracy on this specific test set, what is the key argument for your method? Does the superior representation quality translate to other critical benefits, such as improved robustness, better generalization to out-of-distribution samples, or greater fairness?**
>
> **Ans5:** This is a critical point. While baselines can achieve high predictive accuracy on this single test set, our work is built on the established consensus that **structured representations are essential for solving complex tasks.**
>
> A large body of work confirms this: DFL [5] showed centralized latents improve classification; SimCLR [1] and MoCo [3] use latent-based methods for unsupervised and  self-supervised tasks; and more recently, TSC [4], RankSim [2], and Rank-N-Contrast [6] all demonstrate that representation learning is key to solving imbalance and regression problems.
>
> This prior work proves why representations matter. Our work provides a new solution for how to achieve this in a multi-task setting. The key argument is not only the V-score, but also the **algebraic structure**. This compositional structure is the critical benefit, providing a **interpretable** and **controllable** representation that baselines completely lack.
>
> **W2. Clarity of Results in Table 1: Ans(W2):** Thank you for these excellent presentation suggestions. We agree completely. In the final camera-ready version, we will revise all tables to include bolding for the best results, horizontal lines to visually separate our method from the baselines, and a more detailed caption defining all concept acronyms (e.g., 'az' for azimuth, 'el' for elevation).
>
> **W4. Reproducibility: Ans(W4):** We apologize for this omission. We are fully committed to reproducibility and will release our complete, documented source code upon the paper's acceptance.
>
> References:
>
> [1] Chen, T., Kornblith, S., Norouzi, M., and Hinton, G. (2020). A simple framework for contrastive learning of visual representations. In International conference on machine learning (ICML).
>
> [2] Gong, B., Wang, Y., Feng, J., and Shokoufandeh, A. (2022). Ranksim: A ranking-based similarity model for efficient and effective retrieval. In AAAI Conference on Artificial Intelligence.
>
> [3] He, K., Fan, H., Wu, Y., Xie, S., and Girshick, R. (2020). Momentum contrast for unsupervised visual representation learning. In IEEE/CVF Conference on Computer Vision and Pattern Recognition (CVPR).
>
> [4]  Li, T., Wu, Z., Kan, M.Y., Jiang, Z.H., and Qi, X. (2022). Targeted supervised contrastive learning for long-tailed recognition. In IEEE/CVF Conference on Computer Vision and Pattern Recognition (CVPR).
>
> [5] Wen, Y., Zhang, K., Li, Z., and Qiao, Y. (2016). A discriminative feature learning approach for deep face recognition. In European conference on computer vision (ECCV).
>
> [6] Zha, K., Lin, Z., Chen, J., Wu, K., and Wang, R. (2023). Rank-n-contrast: Learning continuous representations for regression. In IEEE/CVF Conference on Computer Vision and Pattern Recognition (CVPR).

---

### Official Review · Reviewer_FZR9 · 2025-10-30

**Soundness:** 3
**Presentation:** 4
**Contribution:** 3
**Rating:** 6
**Confidence:** 3

**Summary:**

This paper investigates how to reduce conflicting gradients in multi-objective optimization. Rather than modifying gradients during training as with prior methods, this paper proposes to modify the representation space itself instead by allocating task-specific orthogonal subspaces. Specifically, at each training epoch, i) eigenvectors of the features are computed on the entire training data; ii) the top-m eigenvectors are selected as the task domain and used to construct orthogonal subspaces; iii) each latent feature is then projected onto each orthogonal, concept-specific subspace and the sub-space-specific representations are pooled together; and iv) the projected features are then used to compute the training loss.

The authors evaluate their approach on ShapeNet, MPIIGaze, and Rotated MNIST where it outperforms baseline and comparison multi-task methods. They also show through quantitative and qualitative analyses that the representation space created by their method is indeed more structured and composable than those of other methods.

**Strengths:**

1. The proposed approach tackles the problem of multi-objective optimization from a different perspective than prior work.

2. The proposed orthogonal domain is intuitive and well-motivated. The authors illustrate the general problem it should solve and demonstrate through analyses that it is indeed an issue in practice.

3. The domain expansion approach is principled and the authors describe its operations and properties.

4. The authors show that their method outperforms baselines both based on predictive performance and in terms of the quality of the representation space.

**Weaknesses:**

1. Training inefficiency: Eigenvectors need to be calculated and features need to be projected at every epoch. In addition, the Hungarian algorithm needs to be used to align eigenvectors across training epochs. It would be useful to provide some analysis regarding the training times of the proposed method vs the baselines to understand to what extent this is a limitation in practice.

2. Weak base model: The method is only applied to a fairly old model (ResNet-50). It would be helpful to apply the method to more recent models to see whether it can generalize and is still useful with stronger base models.

3. Assumption of orthogonality: The method assumes that concepts should be represented based on orthonormal feature spaces. While this may be optimal when concepts and tasks are clearly disentangled as in the experiments, this may reduce sharing of information when objectives are more closely related as is common in real-world applications (e.g., when viewing language modeling, i.e., next-token prediction as multi-task learning, many words rely on shared latent concepts for their prediction). I would appreciate an analysis (can be in a synthetic setting) that provides some insight regarding whether this is an issue in practice.

4. Narrow evaluation setting: The proposed method is applied to relatively similar datasets containing rotated shapes/digits or gazes. Prior work in this area ([https://arxiv.org/pdf/2001.06782](https://arxiv.org/pdf/2001.06782), [https://arxiv.org/pdf/2010.05874](https://arxiv.org/pdf/2010.05874)) has been evaluated on a broader set of more complex benchmarks including multi-task and multi-label image classification (MultiMNIST, CityScapes, CelebA, multi-task CIFAR-100, NYUv2), multi-task RL, and multi-task NLP datasets. It would be useful to evaluate on more diverse benchmarks to understand the method’s broader applicability and the extent it is applicable to more complex multi-task learning problems.

**Questions:**

1. What is the impact of fitting the eigenvectors on the current batch vs larger subsets of the training data vs the entire training dataset?

2. What is the impact of different choices of the number of eigenvectors M? Is it important for M to be the same as the number of objectives? What happens if M is lower or larger?

3. How does the approach perform with larger sets of concepts (combining Objective Sets 1 and 2)?

---

> ### Author Response · Authors · 2025-11-21
> **Response to Reviewer FZR9 (Part 1)**
>
> We sincerely thank Reviewer FZR9 for their positive feedback and insightful review. We are delighted that you found our approach **"intuitive and well-motivated"**, our method **"principled"**, and our presentation **"excellent"**. Your questions about efficiency, generality, and the orthogonality assumption are all critical, and we have provided new experiments (in **supplementary material**) and clarifications to address each of them.
>
> **W1:Training inefficiency: Eigenvectors need to be calculated and features need to be projected at every epoch. In addition, the Hungarian algorithm needs to be used to align eigenvectors across training epochs. It would be useful to provide some analysis regarding the training times of the proposed method vs the baselines to understand to what extent this is a limitation in practice.**
>
> **Ans1:** Thank you for raising this important practical concern. We analyze this cost in two parts:
>
> 1. **Training**: You are correct that eigendecomposition adds overhead. To precisely quantify this, we ran a new analysis comparing the training time and memory of our method against the baseline. Our experiments were run on a server with an AMD EPYC 7413 24-Core Processor and an NVIDIA A100 GPU. We trained the encoder on Objective Set 1 for 100 epochs with a batch size of 256. The table below shows the average results per epoch. As the results show, our method adds only a manageable computational overhead (approx. 20% increase in epoch time) while having a negligible impact on peak GPU memory. This cost is due to the $O(D^3)$ eigendecomposition on the 2048-D features. **We also note that this eigendecomposition is computed on the CPU, so this 20% time overhead does not increase GPU load**.
>
> | Method | Avg. Epoch Time | Max GPU Memory |
> | :--- | :---: | :---: |
> | Baseline | 1.95 min | 11.2 GB |
> | Ours (Domain Expansion) | 2.34 min | 11.3 GB |
>
> 2. **Inference**: This is a key strength of our approach. After training, the orthogonal basis $V_M$​ is **fixed**. An inference pass only requires the standard encoder forward pass and a simple, computationally lightweight projection. Thus, **inference speed is not affected** and is virtually identical to the baseline.
>
> We believe this manageable, one-time training cost is a reasonable trade-off for the significant gains in interpretability and compositionality.
>
>
> **W2: Weak base model: The method is only applied to a fairly old model (ResNet-50). It would be helpful to apply the method to more recent models to see whether it can generalize and is still useful with stronger base models.**
>
> **Ans2:** This is an excellent point. To test our framework's generality beyond ResNet-50, we first ran a new baseline experiment to see if the objective functions we use (SupCon, RNC) are even compatible with a ViT backbone. We found that running a standard ViT baseline with just the SupCon or RNC losses (i.e., without our Domain Expansion framework) does not converge.
> We are also actively investigating the cause, though we do not have a definitive solution yet. Our initial investigation into the literature suggests this may be a non-trivial and broader challenge. For example, we noted that the original papers for our objective functions, SupCon (Khosla et al., 2020) and Rank-N-Contrast (Zha et al., 2023), do not report results on transformer-based backbones.
>
> We hypothesize that these losses, which were designed for the spatial, global average-pooled features of a CNN, may not be "plug-and-play" compatible with the token representation of a ViT. token architecture.
> Therefore, adapting these specific losses to transformer architectures appears to be an independent and open research problem, which we will highlight as a direction for future work in our discussion.

---

> ### Author Response · Authors · 2025-11-21
> **Response to Reviewer FZR9 (Part 2)**
>
> **W3: Assumption of orthogonality: The method assumes that concepts should be represented based on orthonormal feature spaces. While this may be optimal when concepts and tasks are clearly disentangled as in the experiments, this may reduce sharing of information when objectives are more closely related as is common in real-world applications (e.g., when viewing language modeling, i.e., next-token prediction as multi-task learning, many words rely on shared latent concepts for their prediction). I would appreciate an analysis (can be in a synthetic setting) that provides some insight regarding whether this is an issue in practice.**
>
> **Ans3:** We are grateful for this question, as it touches on the core of our philosophy. First, we would like to clarify that our framework does not assume that concepts and tasks are independent or disentangled. As we showed in our new experiment on redundant tasks (detailed in our answer to Q3 later), correlated concepts can be represented orthogonally in our latent space without a negative impact.
>
> Regarding your key point on "sharing of information when objectives are more closely related" — we absolutely agree with your viewpoint, and it is actually aligned with our research. From our perspective, orthogonality in the latent space does not hurt correlated concepts. Instead, it provides **a structured foundation and a reasonable way to compose this shared information**. To use your excellent example, when a language model relies on shared latent concepts to predict the next word, our framework is theoretically able to provide a reasonable composition for those shared concepts. In addition, the design of this latent composition is **proactive** and **controllable** in our framework.
>
> Though there would certainly be implementation challenges in scaling this to a domain like language, we humbly believe that our core idea of latent construction and concept composition could be a promising direction.
>
>
> **W4: Narrow evaluation setting: The proposed method is applied to relatively similar datasets containing rotated shapes/digits or gazes. Prior work in this area (https://arxiv.org/pdf/2001.06782, https://arxiv.org/pdf/2010.05874) has been evaluated on a broader set of more complex benchmarks including multi-task and multi-label image classification (MultiMNIST, CityScapes, CelebA, multi-task CIFAR-100, NYUv2), multi-task RL, and multi-task NLP datasets. It would be useful to evaluate on more diverse benchmarks to understand the method’s broader applicability and the extent it is applicable to more complex multi-task learning problems.**
>
> **Ans4:** We thank the reviewer for this important point and agree that scaling to more complex benchmarks is a critical next step. For this foundational work, we intentionally chose datasets (ShapeNet, MPIIGaze) that provided **clean, verifiable conceptual axes** (e.s., pose, gaze) . This controlled setting was essential to rigorously demonstrate our paper's core claims: first, that we can **verifiably prevent collapse**, and second, that we can build a functional, **compositional algebra** (Sec. 4.3) , which would be difficult to measure in a more entangled dataset like CityScapes.
>
> We believe this paper provides the necessary proof-of-concept, and we will be sure to highlight scaling to these more complex benchmarks as the primary direction for future work in our discussion.

---

> ### Author Response · Authors · 2025-11-21
> **Response to Reviewer FZR9 (Part 3)**
>
> **Q1: What is the impact of fitting the eigenvectors on the current batch vs larger subsets of the training data vs the entire training dataset?**
>
> **Ans1:** This is a critical point. To be precise: our methodology *always* differentiates between the training mini-batch ($B_{\text{train}}$) and the larger set of samples used for the once-per-epoch covariance estimation ($B_{\text{cov}}$).
> To quantify the numerical stability of this estimation, we ran a new ablation study (**Table 7, Appendix A.6**) to test the sensitivity of performance to the size of $B_{\text{cov}}$. We varied $B_{\text{cov}}$ from 10% to 100% of the total training set.
>
> Our results show that performance is remarkably stable. Even when using only 10% of the training data for the covariance estimation, the performance remains high: **Spearman scores $\approx$  0.9 or higher, V-scores  $>$ 0.9, MAE for regression tasks  $\approx$ 0.08, and classification accuracy $>$ 0.95**. This confirms our method's numerical stability, demonstrating that a stable basis can be estimated efficiently from even a small subset of the data.
>
>
> **Q2: What is the impact of different choices of the number of eigenvectors M? Is it important for M to be the same as the number of objectives? What happens if M is lower or larger?**
>
> **Ans2:** This is a very insightful question. In our current work, we set M equal to the number of objectives, as our goal was to dedicate one axis of variance to each known concept.
>
> If $M$ < num_tasks, multiple concepts would be forced to share a subspace, which would likely re-introduce the representation collapse we aim to solve.
>
> If $M$ > num_tasks, the extra ($M$ - num_tasks) axes would capture additional, unsupervised sources of variance, which is a fascinating avenue for semi-supervised learning.
>
> Your question also highlights a powerful extension: *assigning multiple eigenvectors to a single complex concept* (e.g., representing rotation $\theta$ with two axes for $[\cos\theta, \sin\theta]$). Here, we also share our insight for high-dimensional latent space. When we use one eigenvector to represent a target concept, the latent distribution is a line (vector) in high-dimensional space, which is easy to learn for linear decoders. If we use multiple eigenvectors to represent a concept, the latent distribution becomes a **manifold**. This increases the decoders’ challenge to understand the *meaning* latent distribution. The latent manifold learning and the decoder design is a key part of our future work. We are grateful for your question here.
>
>
> **Q3: How does the approach perform with larger sets of concepts (combining Objective Sets 1 and 2)?**
>
> **Ans3:** This is a crucial question. The reviewer is correct that strict orthogonality might be suboptimal for highly correlated tasks.  To test this, we designed a new experiment where we explicitly introduced a redundant task by duplicating objective set 1 and assigning it to a new orthogonal axis. As shown in **Table 8 (Appendix A.7)**, the model remained highly stable. Despite the added redundant task, the **Spearman and V-scores still averaged above 0.85**. Furthermore, the predictive performance (MAE and Accuracy) was almost identical to the default setting (**MAEs $\approx$ 0.09, Acc. > 0.97**), with only a minor, non-critical drop on a few metrics. Crucially, this performance is still significantly stronger than the baselines in Table 1 (main paper). This result confirms that our framework is stable to task correlation and redundancy.

---

### Official Review · Reviewer_yLyP · 2025-11-02

**Soundness:** 3
**Presentation:** 3
**Contribution:** 3
**Rating:** 6
**Confidence:** 5

**Summary:**

The paper introduces Domain Expansion, a novel framework for multi-task learning (MTL) that addresses latent representation collapse categorized as a failure mode where conflicting task gradients lead to entangled, suboptimal shared representations. Instead of modifying gradient updates, the authors propose orthogonal pooling, an architectural mechanism that explicitly decomposes the latent space into mutually orthogonal subspaces, each dedicated to a specific learning objective. This approach aims to prevent interference between tasks structurally, not procedurally. The paper offers a clear three-stage method: (1) compute latent feature covariance, (2) derive an orthonormal basis via eigendecomposition, and (3) project features into task-specific orthogonal subspaces. Each subspace is decoded separately, and training proceeds with independent task losses. Experiments on ShapeNet, MPIIGaze, and Rotated MNIST demonstrate improvements in representation disentanglement and predictive metrics compared to gradient-based MTL baselines (PCGrad, IMTL, Nash-MTL, FAMO). The authors further show that the resulting latent space supports algebraic concept manipulation, such as vector addition and subtraction of concepts.

**Strengths:**

1. **Novel and intuitive framing:** The notion of latent representation collapse as a geometric phenomenon in shared latent spaces is well-motivated and connects neatly to known issues in MTL such as negative transfer and conflicting gradients.

2. **Architectural elegance:** Orthogonal pooling is conceptually simple yet effective. It shifts focus from gradient-level interventions (as in PCGrad or Nash-MTL) to a proactive representation-level solution.

3. **Compositional algebra:** The inclusion of concept-level operations ($\oplus$, $\ominus$) adds a unique interpretability dimension to MTL. Section 3.3 formalizes this idea cleanly, showing how orthogonal projections allow structured latent arithmetic.

4. **Methodological clarity:** The training pipeline (Fig. 5) and algorithmic description (Eqs. 3 to 8) are logically structured and mathematically well-grounded.

5. **Empirical experiments:** The method outperforms gradient-based MTL baselines on both representation and predictive metrics (Table 1), with qualitative PCA visualizations (Fig. 6) convincingly showing disentanglement.

6. **Interpretability and compositionality:** Demonstrating vector arithmetic over latent subspaces (Sec. 4.3) is original and compelling; it strengthens the claim that the method yields a structured, interpretable representation.

7. **Presentation quality:** The paper is exceptionally well written and illustrated. Figures 1-4 intuitively connect abstract mathematical ideas to geometric and conceptual metaphors (e.g., the “anamorphic art” analogy in Fig. 3 is particularly effective).

**Weaknesses:**

# Major Concerns

1. **Empirical scope is narrow.**
Experiments focus exclusively on relatively small, controlled datasets (ShapeNet, MPIIGaze, Rotated MNIST). These are synthetic or low-dimensional settings. The paper’s claims of scalability and generality (e.g., toward fairness or multimodal learning in Sec. 6) are not yet substantiated.

2. **Unclear stability and computational cost.**
The method requires per-epoch covariance estimation and eigendecomposition of large latent spaces (2048-D ResNet-50 features). This step may be $\mathcal{O}(D^3)$ and unstable for high-dimensional features. The paper should clarify computational cost and numerical stability, especially since Hungarian alignment is used to track basis permutations.

3. **Ambiguity in orthogonality enforcement.**
The orthogonal subspaces are defined using the empirical covariance’s eigenvectors. Thus, orthogonality is not learned but imposed externally. This means the decomposition depends heavily on data statistics rather than explicit task semantics. It’s unclear whether orthogonality aligns with task-specific gradients or merely decorrelates features statistically.

4. **Potential circularity in “concept assignment.”**
The mapping between eigenvectors (principal components) and tasks (concepts) appears heuristic: the top-M eigenvectors are “assigned” to the M objectives. This lacks theoretical grounding. Why should the top eigenvector correspond to azimuth rather than color, for example? A data- or loss-driven criterion would strengthen this link.

5. **Overstated compositionality claims.**
The algebraic operators rely on linear subspace assumptions that may not hold for nonlinear encoders and decoders. Though conceptually elegant, the empirical evidence (Table 1 last column) is limited to cosine similarity metrics. A visual or task-level demonstration (e.g., compositional generation or interpolation) would better support the claim.

6. **Baselines and statistical rigor.**
Results in Table 1 show large performance gaps (e.g., Spearman 0.95 vs 0.49), which seem unusually high. The paper should include standard deviations, repeated trials, and clarification on hyperparameter tuning fairness. Without this, the reported improvements risk appearing overstated.

7. **Orthogonality and task granularity.**
The method presumes one axis per task (Eq. 5), which implicitly assumes tasks are linearly independent. Many real-world MTL setups involve correlated objectives (e.g., segmentation + depth). The framework does not discuss how partially dependent tasks are handled or whether strict orthogonality might hurt shared feature reuse.

8. **Dependence on labeled concept spaces.**
The approach assumes explicit supervision for each “concept” (Sec. 3.3), limiting its applicability to unsupervised or weakly labeled MTL. The contrastive + ranking objectives help, but it remains unclear how Domain Expansion would generalize beyond fully annotated multi-concept datasets.

## Minor Concerns

1. **Equation references:** Eq. (8) could clarify whether projection operators $P_m$ are recomputed each epoch or fixed post-training.

2. **Notation clarity:** Use consistent notation between $f_{proj, m}$ and $F_{proj,m}$
3. **Appendix details:** Appendix A.1 should include pseudocode for the orthogonal pooling operation and basis stabilization via Hungarian alignment.
4. **Baselines:** Including PCGrad (Yu et al., 2020) explicitly in results tables would give a fairer picture since conceptually it is designed to improve multi-task learning dynamics.
5. **Discussion:** Section 5’s “chair $\oplus$ boat” example is charming but anecdotal. Please clarify if such compositions were empirically tested.
6. **Formatting:** Some tables and figure captions (Table 1, Fig. 6) could benefit from indicating whether higher is better for each metric directly in the caption.

**Questions:**

1. **Concept assignment:** How is each eigenvector mapped to a specific task concept? Is there a data-driven matching (e.g., gradient alignment) or manual assignment based on variance ranking?

2. **Computation cost:** What is the training overhead of recomputing eigendecomposition (Eq. 4) per epoch compared to gradient-based MTL baselines?

3. **Dynamic tasks:** Can Domain Expansion adapt if the set of tasks changes mid-training (e.g., new objective added)?

4. **Partial supervision:** Can orthogonal pooling be extended to unsupervised or semi-supervised MTL where some tasks lack labels?

5. **Orthogonality realism:** How sensitive is the method to correlated or redundant tasks? Does strict orthogonality ever hurt performance?

6. **Hungarian alignment:** Why choose Hungarian alignment to track basis order? Would Procrustes or canonical correlation alignment suffice?

7. **Compositional validity:** Beyond cosine similarity, have you tested compositional operators on downstream performance (e.g., predicting unseen concept combinations)?

8. **Numerical stability:** Does covariance estimation introduce instability for large batch or small-batch regimes? Any regularization applied (e.g., shrinkage, clipping)?

9. **Generality:** Can this method be plugged into transformer-based multimodal models (e.g., CLIP, ViT) without modification?

10. **Fair comparison:** Were all baselines tuned to optimal weighting parameters, or did you adopt published defaults?

---

> ### Author Response · Authors · 2025-11-21
> **Response to Reviewer yLyP (Part 1)**
>
> We sincerely thank Reviewer yLyP for their exceptionally thorough, insightful, and constructive review. We are greatly encouraged that you recognized **the novelty of our framing** (latent representation collapse), **the architectural elegance of orthogonal pooling**, and **the originality of our compositional algebra**. Your feedback is invaluable. We have provided clarifications and new experimental results (in the **supplementary material**) to address each of your concerns.
>
> **Q1. Concept assignment: How is each eigenvector mapped to a specific task concept? Is there a data-driven matching (e.g., gradient alignment) or manual assignment based on variance ranking?**
>
> **Ans1:** This is a key question. Our assignment is based on variance ranking (i.e., the top-M eigenvectors are assigned to the $M$ objectives). However, the specific mapping (e.g., whether $v_0$​ is assigned to azimuth or category) is *arbitrary*.
> The reason this works is central to our method: the loss (Eq. 8) is computed after the projection. The gradient for a specific concept, $L_m$​, only backpropagates through its corresponding projected features, $f^{proj, m}$ (which lie on the axis $v_m$​). This training dynamic forces the encoder to learn to map all variance related to concept $C_m$​ onto whichever axis $v_m$ it has been assigned.
>
> To prove this, we ran a new experiment where we randomly shuffled the eigenvector-to-concept assignments and retrained the model. As shown in our **new Table 6 (Appendix A.5)**, the final performance is not negatively impacted. The results **remain high, with Spearman and V-scores at approximately 0.9 and MAE for regression remaining around 0.09**, almost the same as the default setting. This confirms that the model is not relying on a pre-existing alignment of variance; rather, our method creates this alignment during training.
>
> That said, your suggestion of a data-driven matching is an excellent idea for a different problem: extending our work to unsupervised settings where the concepts are not known a priori. We will highlight this as a promising direction for future work.
>
> **Q2:Computation cost: What is the training overhead of recomputing eigendecomposition (Eq. 4) per epoch compared to gradient-based MTL baselines?**
>
> **Ans2:** Thank you for raising this important practical concern. We analyze this cost in two parts:
> 1. **Training**: You are correct that eigendecomposition adds overhead. To precisely quantify this, we ran a new analysis comparing the training time and memory of our method against the baseline. Our experiments were run on a server with an AMD EPYC 7413 24-Core Processor and an NVIDIA A100 GPU. We trained the encoder on Objective Set 1 for 100 epochs with a batch size of 256. The table below shows the average results per epoch. As the results show, our method adds only a manageable computational overhead (approx. 20% increase in epoch time) while having a negligible impact on peak GPU memory. This cost is due to the $O(D^3)$ eigendecomposition on the 2048-D features. **We also note that this eigendecomposition is computed on the CPU, so this 20% time overhead does not increase GPU load**.
>
> | Method | Avg. Epoch Time | Max GPU Memory |
> | :--- | :---: | :---: |
> | Baseline | 1.95 min | 11.2 GB |
> | Ours (Domain Expansion) | 2.34 min | 11.3 GB |
>
> 2. **Inference**: This is a key strength of our approach. After training, the orthogonal basis $V_M$​ is **fixed**. An inference pass only requires the standard encoder forward pass and a simple, computationally lightweight projection. Thus, **inference speed is not affected** and is virtually identical to the baseline.
>
> We believe this manageable, one-time training cost is a reasonable trade-off for the significant gains in interpretability and compositionality.

---

> ### Author Response · Authors · 2025-11-21
> **Response to Reviewer yLyP (Part 2)**
>
> **Q3:Dynamic tasks: Can Domain Expansion adapt if the set of tasks changes mid-training (e.g., new objective added)?**
>
> **Ans3:** This is an excellent question about our framework's flexibility. While our paper's main experiments assume a fixed set of M tasks, we designed a new experiment specifically for this rebuttal to prove that Domain Expansion can be adapted to a continual learning setting. We successfully added N new objectives ($C_{M+1},...,C_{M+N}$​) to a model already pre-trained on M tasks, without retraining from scratch.
>
> Our approach is as follows:
>
> 1. *Freeze Existing Axes:* We "freeze" the original $M$ axes ($v_0​...v_{M−1}$​) and initialize the encoder, which have already been trained.
>
> 2. *Find N New Orthogonal Axes:* We find $N$ new axes by running eigendecomposition on the "residual" feature space $F’$ (where $f^′=f−\sum_{i=0}^{M-1} f^{proj,i}$). We select the top N eigenvectors from this residual space, which ensures they are orthogonal to all previously learned axes.
>
> 3. *Train New Tasks (with Regularization):* We then train the $N$ new tasks. Each new objective $C_{M+j}$​ is trained using its loss $L_{M+j}$​ on projections onto its corresponding new axis $v_{M+j−1}$​
>
> 4. *Prevent Catastrophic Forgetting:* To prevent the encoder from "forgetting" the original tasks (i.e., catastrophic forgetting), we add an L2 regularization loss. This loss ensures that the projected coefficients for the original $M$ tasks (on their frozen axes $v_0​...v_{M−1}$​) remain constant while the $N$ new tasks are being learned.
>
> The total loss function for this new training phase is defined *eq.(25) Appendix A.4*
>
> The results of this experiment are presented in **Table 5, Appendix A.4**. The "Continual" model, which was fine-tuned, shows a **minor** performance drop compared to the "Default" model trained on all tasks from scratch. This drop also affects the concept composition quality, which decreases from 0.95 to 0.72. However, the framework still successfully **solves the latent collapse problem** and **prevents catastrophic forgetting**. Crucially, even with this degradation, the model's performance on both representation (e.g., Spearman, V-score > **0.8**) and compositionality (**0.72**) remains remarkably stronger than the baselines in Table 1 (main paper), demonstrating the framework's viability for continual learning.
>
> **Q4:Partial supervision: Can orthogonal pooling be extended to unsupervised or semi-supervised MTL where some tasks lack labels?**
>
> **Ans4:** This is an insightful point. Our orthogonal pooling mechanism is agnostic to the specific loss functions used in each subspace. In this work, we used supervised losses (RNC, SupCon) because we had labeled concepts. However, the framework could readily be extended to unsupervised or semi-supervised settings by applying different losses to the projected features $F^{proj, m}$​. For example, unsupervised and self-supervised objectives (like MoCo [1] or SimCLR [2]) could be used to structure one subspace, while a supervised loss operates on another. This is a key part of our future work, as noted in Q1.
>
> References:
>
> [1] He, Kaiming, et al. *Momentum Contrast for Unsupervised Visual Representation Learning.* Proceedings of the IEEE/CVF Conference on Computer Vision and Pattern Recognition (CVPR), 2020, pp. 9729–9738.
>
> [2] Chen, Ting, et al. *A Simple Framework for Contrastive Learning of Visual Representations.* Proceedings of the 37th International Conference on Machine Learning (ICML), 2020, pp. 1597–1607.

---

> ### Author Response · Authors · 2025-11-21
> **Response to Reviewer yLyP (Part 3)**
>
> **Q5:Orthogonality realism: How sensitive is the method to correlated or redundant tasks? Does strict orthogonality ever hurt performance?**
>
> **Ans5:** This is a crucial question. The reviewer is correct that strict orthogonality might be suboptimal for highly correlated tasks.  To test this, we designed a new experiment where we explicitly introduced a redundant task by duplicating objective set 1 and assigning it to new orthogonal axes. As shown in **Table 8 (Appendix A.7)**, the model remained highly stable. Despite the added redundant task, the **Spearman and V-scores still averaged above 0.85**. Furthermore, the predictive performance (MAE and Accuracy) was almost **identical to the default setting (MAEs $\approx$ 0.09, Acc. > 0.97)**, with only a minor, non-critical drop on a few metrics. Crucially, this performance is still significantly stronger than the baselines in *Table 1 (main paper)*. This result confirms that our framework is stable to task correlation and redundancy.
>
> **Q6:Hungarian alignment: Why choose Hungarian alignment to track basis order? Would Procrustes or canonical correlation alignment suffice?**
>
> **Ans6:** Thank you for the excellent suggestions. We chose Hungarian alignment to stabilize the basis order during training, following a similar procedure in recent work (e.g., TSC [3]) . As shown in **Appendix A.2**, the analysis shows the basis is adaptive in early training but stably converges, **approaching 1.0 similarity around epoch 60 and reaching **0.99** by epoch 100**, which confirms the stability of Hungarian alignment. We did not explore alternatives like Procrustes or CCA, but we agree these are valid methods for alignment and represent a potential area for future technical refinement.
>
> **Q7:Compositional validity: Beyond cosine similarity, have you tested compositional operators on downstream performance (e.g., predicting unseen concept combinations)?**
>
> **Ans7:** This is an excellent suggestion. We agree that testing on unseen concept combinations is a powerful method for evaluating compositional generalization.
> Our current experiment in Table 1 (main paper) is the direct test for the prerequisite: it proves that the latent space has learned the correct geometric structure for vector arithmetic. We show that the result of an algebraic operation geometrically matches the vector of the expected combined concept.
>
> The reviewer's suggested experiment is a valuable but non-trivial next step, as it requires a new data-splitting and training protocol. Unfortunately, we were unable to complete this complex new experiment within the limited time and computing resources of the rebuttal period. We thank the reviewer for the insightful suggestion and will add this as a key direction for future work.
>
> **Q8:Numerical stability: Does covariance estimation introduce instability for large batch or small-batch regimes? Any regularization applied (e.g., shrinkage, clipping)?**
>
> **Ans8:** This is a critical point. To be precise: our methodology always differentiates between the training mini-batch ($B_{\text{train}}$) and the larger set of samples used for the once-per-epoch covariance estimation ($B_{\text{cov}}$).
> To quantify the numerical stability and efficiency of this estimation, we ran a new ablation study (**Table 7, Appendix A.6**) to test the sensitivity of performance to the size of $B_{\text{cov}}$. We varied $B_{\text{cov}}$ from 10% to 100% of the total training set.
>
> Our results show that performance is remarkably stable. Even when using only 10% of the training data for the covariance estimation, the performance remains high: **Spearman scores are $\approx$ 0.9 or higher, V-scores are $>$ 0.95, MAE for regression tasks is $\approx$ 0.08, and classification accuracy is $>$ 0.95**. This confirms our method's numerical stability, demonstrating that a stable basis can be estimated efficiently from even a small subset of the data.
>
> References:
>
> [3] Li, Tianhong, et al. "Targeted Supervised Contrastive Learning for Long-Tailed Recognition." Proceedings of the IEEE/CVF Conference on Computer Vision and Pattern Recognition (CVPR), 2022, pp. 6918–6928. IEEE, https://doi.org/10.1109/CVPR52688.2022.00679.

---

> ### Author Response · Authors · 2025-11-21
> **Response to Reviewer yLyP (Part 4)**
>
> **Q9:Generality: Can this method be plugged into transformer-based multimodal models (e.g., CLIP, ViT) without modification?**
>
> **Ans9:** This is a key question. To answer it, we first ran a new baseline experiment to see if the objective functions we use (SupCon, RNC) are even compatible with a ViT backbone.
> We found that running a standard ViT baseline with just the SupCon or RNC losses (i.e., without our Domain Expansion framework) does not converge.
>
> We are also actively investigating the cause, though we do not have a definitive solution yet. Our initial investigation into the literature suggests this may be a non-trivial and broader challenge. For example, we noted that the original papers for our objective functions, SupCon (Khosla et al., 2020) and Rank-N-Contrast (Zha et al., 2023), do not report results on transformer-based backbones.
>
> We hypothesize that these losses, which were designed for the spatial, global average-pooled features of a CNN, may not be "plug-and-play" compatible with the token representation of a ViT. token architecture.
> Therefore, adapting these specific losses to transformer architectures appears to be an independent and open research problem, which we will highlight as a direction for future work in our discussion.
>
> **Q10:Fair comparison: Were all baselines tuned to optimal weighting parameters, or did you adopt published defaults**
>
> **Ans10:** We appreciate the need for fair tuning. For our main experiments, we did not simply use published defaults, as these are not optimal for this specific combination of tasks.
>
> Instead, for all methods evaluated—including the simple baseline, the gradient-based MTL baselines (FAMO, Nash-MTL, IMTL), and our own—we tuned the loss weights $w_m$​ to ensure the magnitude of each loss component was balanced. For example, the scale of SupCon is generally larger than 200.0 but RNC is around 5.0 in our experiments. Thus, we weigh SupCon with 0.02 to balance the losses' contribution.
>
> This ensures that no single task's loss dominates the optimization and provides the most favorable and robust setup for every method. The significant performance gaps reported in Table 1 (main paper) are therefore not an artifact of poor baseline tuning, but rather reflect the fundamental differences in how each method handles the multi-objective conflict.

---

> ### Comment · Reviewer_yLyP · 2025-11-24
> **Requesting Clarifications**
>
> Thank you for putting together a through rebuttal. Most of my concerns were addressed, and I seek few clarifications.
>
> 1. **Regarding Continual Learning Experiment**: If I understand correctly, the "Default" model in Table 5 has been trained on M+N tasks from scratch, whereas the continual model is fine-tuned on N tasks after being pre-trained on M tasks. What I find amusing is, if the M subspaces are already orthogonal, introducing N new orthogonal subspaces should not effect the geometric structure of the latent spaces for pre-trained tasks. Because, these N new subspaces are orthogonal to the previous M subspaces. The main contribution of this paper is to exploit orthogonality of subspaces to improve MTL. Then why we see a performance drop in Table 5 on pre-training tasks for the "Continual Model"? What am I missing here? Why is the orthogonality not working? Could you provide the mean and standard deviation of the performance measured over 5 different random seeds? It is unclear if the performance drop is statistically significant or not.
>
> 2. **Regarding Runtimes**: Could you provide **batch runtime** for training and inference, measured over 10 batches after few warmup epochs, and report mean and standard deviation for both baseline and your method?
>
> 3. **PCGrad**: Why are you not comparing with PCGrad? It is a valid baseline, because PCGrad mitigates task inference/gradient conflicts using projections.

---

> ### Author Response · Authors · 2025-11-27
> **Response to Reviewer yLyP (Part 5)**
>
> **Added Q1:Regarding Continual Learning Experiment: If I understand correctly, the "Default" model in Table 5 has been trained on M+N tasks from scratch, whereas the continual model is fine-tuned on N tasks after being pre-trained on M tasks. What I find amusing is, if the M subspaces are already orthogonal, introducing N new orthogonal subspaces should not effect the geometric structure of the latent spaces for pre-trained tasks. Because, these N new subspaces are orthogonal to the previous M subspaces. The main contribution of this paper is to exploit orthogonality of subspaces to improve MTL. Then why we see a performance drop in Table 5 on pre-training tasks for the "Continual Model"? What am I missing here? Why is the orthogonality not working? Could you provide the mean and standard deviation of the performance measured over 5 different random seeds? It is unclear if the performance drop is statistically significant or not.**
>
> **Ans1:** This is an extremely sharp and insightful question, and we thank the reviewer for it.
>
> To explain precisely what is happening, the 2D analogy you proposed is the clearest method. Let's use that exact example:
> 1. **Initial State:** Assume our latent space is 2D (X, Y) and we have three features: f1(1, 1), f2(2, 3), f3(4, 5).
>
> 2. **Pre-training (M tasks):** We first train the $M$ tasks on the X-axis. The X-axis ($v_X$) and its decoders are now frozen.
>
> 3. **Continual Prep (N tasks):** To add $N$ new tasks, we find a new axis. We compute the residual features by subtracting the (frozen) X-projection:
> f1_resid = (0, 1)
> f2_resid = (0, 3)
> f3_resid = (0, 5)
> We run eigendecomposition on these residuals to find the new Y-axis ($v_Y$). By construction, $v_Y$ is orthogonal to X-direction. We then freeze the $v_Y$ axis.
>
> 4. **Continual Training:** Now, we train the $N$ new tasks on the $v_Y$ axis. Critically, the encoder is NOT frozen. The loss for the $N$ tasks only cares about the Y-values (e.g., it pushes the encoder to make f1's Y-value equal 1). The encoder is free to update the feature in any way to satisfy this.
> It could update f1(1, 1) to f1_new(1 + $\delta_1$, 1).
> It could update f2(2, 3) to f2_new(2 + $\delta_2$, 3).
>
> This $\delta$ shift is **catastrophic forgetting.** The updated feature $f_{new}$ now has the wrong projection on the *original, frozen* X-axis, which causes the performance drop.
>
> To be very precise: we guarantee the **new axis ($v_Y$)** is orthogonal to the old axis X-axis, but we **cannot** guarantee the **updated features** ($f_{new}$) will have the same projection on the old axis.
>
> This is exactly why the L2 regularization term (Eq. 25) is necessary. It is a **soft constraint** designed to minimize that $\delta$ shift, not a hard constraint (like freezing the encoder) to prevent it.
>
> The performance drop (e.g., Spearman/V-score dropping $\approx$0.09 on average for the original tasks) is the measured "cost" of this soft constraint. We see this result not as a failure, but as a **promising trade-off**. A standard baseline, when fine-tuned, would suffer total catastrophic forgetting (a massive $\delta$ shift). Our framework, by explicitly regularizing the projections on the frozen axes, proves to be highly robust in managing this trade-off.
>
> As a final note comparing the two models: the "Default" model (trained on M+N tasks from scratch) has the advantage of optimizing all M+N axes simultaneously within the entire latent space to find a global optimum. In contrast, our "Continual" model **freezes the first M axes**. This forces the new N tasks to find their axes only within the remaining **residual latent space** (which is, by construction, orthogonal to the first M).  This residual space is inherently more constrained, offering less flexibility. This structural constraint, in addition to the small $\delta$ shift, also helps explain the performance gap relative to the "Default" model.
>
> The results, presented below, confirm a small, consistent drop in performance. This is the expected outcome and aligns with our theoretical $\delta$ shift explanation. This data supports our analysis that the drop is not a random failure, but is the measured and expected trade-off for maintaining plasticity while regularizing the original tasks. We thank the reviewer for this suggestion, as it strengthens our paper's analysis.
>
>
> | Metric | Task az (Spearman) | Task el (Spearman) | Task rot (Spearman) | Task cls (V-score) | Task id (V-score)|
> | :--- | :---: | :---: | :---: | :---: | :---: |
> | Score | 0.90 ± 0.04 | 0.78 ± 0.04 | 0.81 ± 0.03 | 0.77 ± 0.09 | 0.77 ± 0.07|
>
> | Metric | Task az (MAE) | Task el (MAE) | Task rot (MAE) | Task cls (Acc.) | Task id (Acc.)| Comp.
> | :--- | :---: | :---: | :---: | :---: | :---: | :---: |
> | Score | 0.14 ± 0.04 | 0.12 ± 0.02 | 0.12 ± 0.01 | 0.96 ± 0.04 | 0.93 ± 0.08| 0.57 ± 0.09

---

> ### Author Response · Authors · 2025-11-27
> **Response to Reviewer yLyP (Part 6)**
>
> **Added Q2:Regarding Runtimes: Could you provide batch runtime for training and inference, measured over 10 batches after few warmup epochs, and report mean and standard deviation for both baseline and your method?**
>
> **Ans2:** We are very grateful to the reviewer for this excellent suggestion to perform a more precise micro-benchmark. This new data clarifies our cost analysis significantly.
>
> We ran the experiment exactly as requested, measuring the per-batch runtime (in seconds) over 20 consecutive batches after 5 warmup epochs.
>
>
> | Training | Baselines | Ours |
> | :--- | :---: | :---: |
> | batch time | 1.36 ± 0.63 sec | 1.49 ± 2.09 sec |
>
> | Inference | Baselines | Ours |
> | :--- | :---: | :---: |
> | batch time | 0.15 ± 0.36 sec | 0.11 ± 0.15 sec |
>
> As this data shows, the mean per-batch overhead for training is 0.13 seconds (1.49 - 1.36), which is a $\approx$ 9.5% increase. This new, more precise benchmark helps clarify the "epoch time" cost reported in our other analyses. It suggests that the overhead is a combination of our per-batch operations and the once-per-epoch eigendecomposition.
> We believe this is a very favorable and manageable cost profile, as the per-batch cost is not prohibitive.
>
>
> **Added Q3:PCGrad: Why are you not comparing with PCGrad? It is a valid baseline, because PCGrad mitigates task inference/gradient conflicts using projections.**
>
> **Ans3:** Our reasoning for not including it as a direct baseline was that the current methods (FAMO, Nash-MTL, and IMTL) we do compare against are widely recognized in the literature as outperforming PCGrad.
>
> Therefore, we made the deliberate choice to benchmark our method against these more **recent** and **challenging** baselines. Since our method demonstrates a clear improvement over these methods, we believe this transitively shows an advantage over PCGrad as well.

---

### Comment · Area_Chair_LA7f · 2025-11-27
**Request for Timely Response to Authors’ Rebuttal and Discussion**

Dear Reviewers,

I hope you are doing well. The authors have now submitted their rebuttal for the paper under your review. At this stage, your timely response is essential for ensuring a smooth discussion phase.

Could you please review the rebuttal at your earliest convenience and share your updated thoughts? If there are points that require further discussion among the reviewers, please feel free to initiate or join the conversation on the discussion thread.

Your prompt input will greatly help us maintain the review timeline. Thank you very much for your efforts and valuable contributions.

Best regards,

AC

---

### Meta-Review · Area_Chair_aC7h · 2025-12-20

**Summary:**

The paper proposes a framework for multi-task learning (MTL) to address the latent representation collapse caused by conflicting gradients.  The paper proposes orthogonal pooling, a mechanism to decompose the latent space into orthogonal subspaces each dedicated to different objectives

Overall, the main thread seems to be the limited evaluation and lack of theoretical backing on the orthogonality assumptions and results.  The authors maintain that their experiments are sufficient to validate the idea which I agree with.  While having more experiments will be interesting and will strengthen the paper, the proposal and the demonstration on a controlled environment should suffice.  Regarding the lack of theoretical support, the authors showed more experiments to support their claims but no theoretical guarantees or backing is provided.  The discussion about the success of existing pre-trained models in multi-objectives (so called "foundation" models by reviewer wQsh) is not totally convincing.  However, under the authors setup this objective is sound, and shows improvements.

Thus, I recommend the acceptance of the paper; but given the limited support to modern architectures (attention-based), empirical evidence provided, and the limited discussion w.r.t. existing multi-objective training that doesn't collapse, I wouldn't mind the rejection of the paper either.

Strengths:

- The motivation behind the geometric interpretation of the latent representation collapse is well explained, and it is different from previous works
- The orthogonal pooling is simple yet effective, and the compositional algebra improves the structured latent interpretation
- The experimental results are convincing for representations and predictive tasks
- Clarity in the presentation

Weaknesses:
- Experiments on small controlled datasets that do not support the claims of scalability and generality
- It is not clear whether the orthogonal subspaces align with the tasks
- The eigenvectors and the tasks relation is not grounded
- The compositionality claims do not hold for the non-linear encoder/decoders used
- Statistical rigor on experiments?
- Dependence on labeled concepts (yet claims of unsupervised learning)
- Weak base models (based on ResNet50) and it doesn't work on ViTs

**Reviewer Concerns:**

Reviewer yLyP is concerned that the experiments do not support the claims (scalability and generality), that the orthogonal subspaces may not align with the tasks and the guarantees on the alignment of the eigenvectors and the tasks, as well as the compositionality. The authors replied to the reviewer concerns and did new experiments to support their claims.  The compositional validity is still questionable given that the authors couldn't provide any supporting theory or experiments.  Similarly, the issues of extending the method to ViT is sill open given that it doesn't work as is. During the rebuttal, the reviewer raised concerns about continual learning experiments and the drop in accuracy.  The authors replied mentioning that it was regarding a trade-off on the encoder given the optimization on the constrained remaining directions.


Reviewer FZR9 is concerned about the overhead imposed by the eigendecomposition, the weak base models, the assumption of orthogonality and the overall narrow evaluation.  The authors answered the reviewer's questions where the limitation on working on ViTs remained opened, and the narrow experiments were justified as proof of concept.

Reviewer CqRr is concerned about the scope of the evaluation given the limited architectural and dataset diversity, readability of the results in Table 1, raised concerns about the practical trade-offs given the lack of diverse metrics.  The replied to the reviewer's questions claiming that their model could generalize but it is beyond this paper, and explain how transferability could work.  However, the main weaknesses remain unaddressed.

Reviewer wQsh raised concerns about previous work using latent spaces to represent concepts, the limited empirical validation given the broad claims, lack of grounding for stability and convergence of the orthogonal basis extraction, and a missing overhead discussion.  The authors replied to the comparison against previous work pointing out that their work is constructive in contrast to existing work that is primarily observational.  Regarding the stability, the authors provide new experiments that support the stability claims.  For the reduced scope, the authors support the selection as these datasets provide controlled experiments.

The reviewer wQsh commented again with concerns regarding the missing complex tasks which weakens the validity of the method, the lack of discussion against "Foundation" Models that work in multi-objective settings, and the lack of discussion against the literature that show the vectors being already orthogonal in the high dimensional setup.  The authors maintain that the experiments are sufficient.  Regarding the orthogonality, the authors claim that their proposal is constructive and that having orthogonal vectors doesn't avoid the collapse in structured losses.

**Reviewer Scores:**

Reviewer CqRr recommended a WR.  The authors replied to the reviewer's comments and argue their stance.  Similar to the previous reviewers, the backbones and the evaluation settings remained contended points.  While the concerns from the reviewer are valid, the authors already present results that validate their claims.  Thus, I weight the concerns less, and value the raised strengths by CqRr (originality and elegance of the method, and the interpretability and compositionality) more.

Reviewer wQsh recommended a R. The authors replied to the reviewer's concerns, and most of the concerns were solved.  The reviewer then raised issues with the insufficient evidence on complex tasks, contradicting claims with existing works on pre-trained multi-objective models ("foundation" models), and orthogonal construction as a necessity.  The authors replied to the different losses used in the pre-trained multi-objective models and the ones they used which differ, and claim that the validation on complex tasks is out of scope.  The orthogonality assumption was supported as a constructive method in contrast to the existing methods.  While the concerns are valid, I believe the authors provided sufficient evidence to support their claims, and the remaining concerns are desirable (e.g., extra experiments in complex tasks) but not fundamental.  I would weight the originality and novelty of the method more than the expected experiments.

Given the open concerns on the weak backbones and limited evaluations, I wouldn't mind to see the paper rejected.  However, I believe the authors provided sufficient evidence to support their claims, and thus I recommend the acceptance of the paper.

---

### Decision · Program_Chairs · 2026-01-26

Accept (Poster)